# Inhibition in the auditory brainstem enhances signal representation and regulates gain in complex acoustic environments

**Christian Keine[1]\*‡, Rudolf Rübsamen[1]†, Bernhard Englitz[2]†**

[1]Faculty of Bioscience, Pharmacy and Psychology, University of Leipzig, Leipzig, Germany; [2]Department of Neurophysiology, Donders Center for Neuroscience, Radboud University, Nijmegen, Netherlands

**Abstract** Inhibition plays a crucial role in neural signal processing, shaping and limiting responses. In the auditory system, inhibition already modulates second order neurons in the cochlear nucleus, e.g. spherical bushy cells (SBCs). While the physiological basis of inhibition and excitation is well described, their functional interaction in signal processing remains elusive. Using a combination of in vivo loose-patch recordings, iontophoretic drug application, and detailed signal analysis in the Mongolian Gerbil, we demonstrate that inhibition is widely co-tuned with excitation, and leads only to minor sharpening of the spectral response properties. Combinations of complex stimuli and neuronal input-output analysis based on spectrotemporal receptive fields revealed inhibition to render the neuronal output temporally sparser and more reproducible than the input. Overall, inhibition plays a central role in improving the temporal response fidelity of SBCs across a wide range of input intensities and thereby provides the basis for high-fidelity signal processing.

**\*For correspondence:** christian.keine@gmail.com

†These authors contributed equally to this work

**Present address:** ‡Molecular Mechanisms of Synaptic Function, Max Planck Florida Institute for Neuroscience, Jupiter, United States

**Competing interests:** The authors declare that no competing interests exist.

## Introduction

Dynamic processing in neural networks is controlled by an interplay of excitation and inhibition. In cortical processing, the dominant excitatory neurons interact reciprocally with inhibitory neurons, which serve key functions in shaping the responses (reviewed in *Isaacson and Scanziani, 2011*). In the auditory cortex, recent work has emphasized the role of inhibition in dynamically balancing excitation via a high degree of co-tuning (e.g. *Wehr and Zador, 2003*; *Renart et al., 2010*) that serves to shape and accelerate network dynamics. Similarly, in other modalities, inhibition was found to be co-tuned with excitation in the cortex, typically with a wider tuning, generating the well-described inhibitory sidebands (auditory: *Wang et al., 2002*; *Wu et al., 2008*, visual: *Sohya et al., 2007*; *Niell and Stryker, 2008*; *Liu et al., 2009*, *2011*; *Katzner et al., 2011*, olfactory: *Poo and Isaacson, 2009*). Temporally, inhibition often follows excitation closely (auditory: *Wehr and Zador, 2003*, somatosensory: *Wilent and Contreras, 2004*).

In the auditory brainstem, the role of inhibition has also been studied, however, from a more fundamental perspective, without a focus on its functional role during complex stimulation. Various studies have shown prominent inhibitory influences on signal processing in the cochlear nucleus (*Caspary et al., 1994*; *Kopp-Scheinpflug et al., 2002*; *Gai and Carney, 2008*), in the medial and lateral superior olive (*Grothe and Sanes, 1993*; *Brand et al., 2002*; *Myoga et al., 2014*), and in the dorsal and ventral nuclei of the lateral lemniscus (*Yang and Pollak, 1994*, *1998*; *Burger and Pollak, 2001*; *Nayagam et al., 2005*; *Pecka et al., 2007*; *Spencer et al., 2015*).

**eLife digest** In humans and other animals, small differences in the time at which a sound arrives at each ear are crucial for determining the location of the sound. Neurons in the first processing station of the brain – the cochlear nucleus – receive information about sounds (or "inputs") from the ears. They then produce electrical signals that relay this information to other areas of the brain. Some of these inputs increase the activity of the neurons and so are known as "excitatory" inputs, while other "inhibitory" inputs decrease the activity of the neurons. The balance between these two inputs determines what information is passed to other parts of the brain, but it is not clear how these inputs interact.

Keine, Rübsamen and Englitz studied electrical activity in the brains of Mongolian gerbils while being exposed to sounds with more natural properties than previously studied. The experiments reveal that inhibitory inputs play an important role in controlling the activity of neurons in the cochlear nucleus. By decreasing the neurons' activity, inhibitory inputs allow these cells to respond to many different levels of sound, from very loud to very quiet. The experiments also show that excitatory and inhibitory inputs are triggered by similar sounds so that the two processes quickly balance each other. This means that the brain is equally able to work out where a sound is coming from regardless of whether it is loud or quiet.

Further work is now needed to understand responses to natural sounds and to determine how experimentally removing the inhibitory inputs affects hearing.

The cochlear nucleus (CN), the first stage of the central auditory system, is the starting point of distinct neuronal circuits involved in sound source localization. Spherical bushy cells (SBC) in the anteroventral division of the CN (AVCN) provide the temporally precise excitatory inputs to binaural neurons in the medial superior olive (MSO), where interaural time differences are computed (*Yin and Chan, 1990*). These SBCs receive suprathreshold excitatory input from auditory nerve fibers (ANF) via large axosomatic terminals, the endbulbs of Held (*Brawer and Morest, 1975*; *Schwartz and Gulley, 1978*; *Ryugo and Sento, 1991*; *Nicol and Walmsley, 2002*). In addition, inhibitory inputs on SBCs have been reported, which provide surprisingly slow acoustically evoked inhibition mediated by glycine and GABA (*Wu and Oertel, 1986*; *Kolston et al., 1992*; *Juiz et al., 1996*; *Lim et al., 2000*; *Mahendrasingam et al., 2004*; *Xie and Manis, 2013*), with glycine dominating (*Nerlich et al., 2014b*).

Due to the requirements of high-fidelity acoustic processing underlying sound localization, many studies focused on the fast and temporally precise signal transmission in auditory brainstem circuits. With respect to changes in temporal precision from ANF to neurons in the CN some studies – comparing population data – reported a general increase in temporal precision (*Joris et al., 1994a*, *1994b*), while others found no change (*Bourk, 1976*; *Blackburn and Sachs, 1989*; *Winter and Palmer, 1990*), or reported decreased temporal precision at certain stimulation frequencies (*Paolini et al., 2001*; *Fukui et al., 2006*).

More recent studies advanced the analysis to the single-cell level, by comparing the endbulb of Held evoked excitatory postsynaptic potentials (EPSP) with the action potentials (AP) of the SBCs allowing for a direct comparison of ANF input and SBC output (*Typlt et al., 2010*). This enabled a direct assessment of the input-output function under the condition of acoustic stimulation, also in combination with pharmacological manipulations. The respective experiments revealed a slight increase in temporal precision of signal coding, attributed to the influence of acoustically evoked inhibition (*Dehmel et al., 2010*; *Kuenzel et al., 2011*; *Keine and Rübsamen, 2015*). It may be argued that the stimulus conditions employed were rather static and did not adequately reflect the challenge of processing the dynamics of spectrotemporal complex acoustic signals. While fast inhibition in T stellate cells has been attributed to a role in comodulation masking release (*Pressnitzer et al., 2001*), the inhibitory dynamics in SBCs seem to be too slow for such an effect (*Xie and Manis, 2013*). Previous studies investigated inhibition using pure tone stimulation, and this is why the functional role of inhibition in signal processing at the ANF-SBC synapse during complex acoustic stimulation has not been fully resolved.

In the present study, we set out to elucidate the functional role of acoustically evoked inhibition at the ANF-SBC synapse using combined in vivo loose-patch recordings with direct iontophoretic manipulation of inhibitory receptors and a detailed input-output signal analysis based on spectro-temporal receptive fields in responses to complex acoustic stimulation. Our results indicate a reliable co-tuning of inhibition with the main excitatory input. While we observed some sharpening of the response in time and frequency, our results suggest that inhibition functions as a gain control that renders the postsynaptic response sparser in time and more reproducible across trials. Temporal sparsity, i.e. a response restricted to fewer time-points, can increase the information per spike while reducing the energy expenditure. Reproducibility, i.e. a more consistent response to identical stimuli, can provide reliable stimulus encoding.

These improvements are a consequence of the combined subtractive/divisive action of glycine (*Kuenzel et al., 2011*, *2015*): The subtractive component enhances the temporal sparsity by raising the threshold for spiking. The divisive component acts primarily as a gain control, which - in conjunction with the co-tuning - maintains the SBC output rate in a smaller range across different stimulus levels. Together these two effects focus the SBC output onto well-timed stimulus events across a wide range of stimulus levels. Thus, inhibition improves the basis for the high-fidelity signal processing in downstream nuclei crucial for sound localization irrespective of the prevailing stimulus levels.

## Results

The interaction between acoustically evoked excitation and inhibition is a key constituent at the initial stages of signal processing in the auditory brainstem (*Kopp-Scheinpflug et al., 2002*; *Dehmel et al., 2010*; *Kuenzel et al., 2011*; *Keine and Rübsamen, 2015*). This study aimed for an investigation of sound-evoked inhibition on the processing of complex structured signals (mimicking broadband acoustic conditions) at the auditory nerve-to-spherical bushy cell synapse (ANF-SBC). A total of 85 units were recorded from the rostral pole of the anteroventral cochlear nucleus (AVCN), the location of large, low-frequency coding SBCs (*Bazwinsky et al., 2008*). The identification of SBCs was based on the following physiological properties: a discernible prepotential in addition to the complex waveform (*Pfeiffer, 1966*; *Englitz et al., 2009*; *Typlt et al., 2010*), short AP duration (*Typlt et al., 2012*), high spontaneous firing rates (*Smith et al., 1993*), and the primary like response pattern to pure-tone stimulation (*Blackburn and Sachs, 1989*). From these 85 cells, 23 were recorded while simultaneously applying glycine receptor agonists and antagonists. Units had a characteristic frequency (CF) of (mean $\pm$ standard deviation) $2.1 \pm 0.6$ kHz and a minimal threshold of (median [first quartile, third quartile]) 7.5 [0.8, 14.9] dB SPL.

To understand how acoustically evoked inhibition shapes SBC output, the present report focuses on the differential analysis between SBC EPSPs that trigger a postsynaptic AP, i.e. $EPSP_{succ}$ and EPSPs that fail to trigger an AP, i.e. $EPSP_{fail}$. Previous studies showed that during spontaneous activity, EPSP amplitudes are close to threshold, such that not all ANF input spikes trigger an SBC output spike. Also, acoustically evoked inhibition interacts dynamically with the EPSPs and prevents output spikes (*Kuenzel et al., 2011*; *Keine and Rübsamen, 2015*).

The respective differences between ANF input and SBC output can be analyzed from the complex waveform of SBC signals consisting of the presynaptic action potential (prepotential, PP) and the excitatory postsynaptic potential (EPSP) which may or may not be followed by an AP (*Figure 1A*). The fast EPSP rising slope served for the detection of both types of signals, while the dynamics of the signals' falling slopes reliably allowed to distinguish between the two: (i) $EPSP_{succ}$, i.e. EPSPs that successfully trigger postsynaptic APs, and (ii) $EPSP_{fail}$, i.e. EPSPs that fail to trigger APs. The maximum falling slope was consistently higher in $EPSP_{succ}$ than in $EPSP_{fail}$ ($EPSP_{succ}$ = $21.1 \pm 4.9$ vs. $EPSP_{fail}$ = $4.4 \pm 1.2$ V/s, difference [$\Delta$] = $16.8 \pm 4.3$ V/s, p<0.001, paired $t$-test, n = 62, U1 = 1, *Figure 1B* left, see also *Figure 1—source data 1*). The sum of $EPSP_{fail}$ and $EPSP_{succ}$ was defined as the ANF input to the SBC, while the subset of $EPSP_{succ}$ indicated the output ascending to the next level of processing, i.e. the superior olivary complex.

Unlike the falling slopes of the signals, the maximal EPSP rising slopes showed considerable overlap between $EPSP_{fail}$ and $EPSP_{succ}$ (*Figure 1B* middle and right). Still, $EPSP_{succ}$ had higher average EPSP rising slopes than $EPSP_{fail}$ ($EPSP_{succ}$ = $9.9 \pm 2.2$ V/s vs. $EPSP_{fail}$ = $7.5 \pm 2.2$ V/s, $\Delta$ = $2.4 \pm 1.5$ V/s, p<0.001, paired $t$-test, n = 62, U1 = 0.1). Considering this difference, the EPSP rising slope can – to some degree – predict the probability of AP generation. During spontaneous

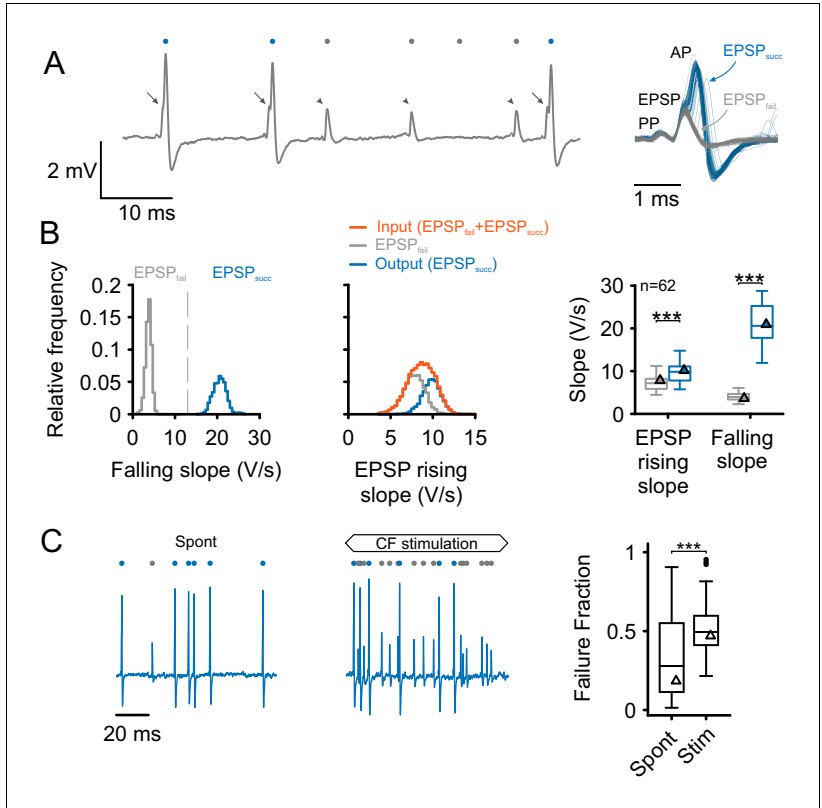

**Figure 1.** Separation and attribution of pre- and postsynaptic neuronal response components. (**A**) Left: Representative trace of an in-vivo loose-patch recording of a spherical bushy cell (SBC) showing both EPSPs followed by an action potential (arrows, blue dots) and EPSPs which fail to trigger an AP (arrowheads, gray dots). Right: Superimposing the events (50 events of each type) shows that both signal types share the presence of a prepotential (PP) and an EPSP, but may (EPSP$_{succ}$) or may not (EPSP$_{fail}$) trigger a postsynaptic AP. (**B**) Left: Both types of events are clearly separable by the maximum falling slope, with APs showing much steeper falling slopes (blue, EPSP$_{succ}$) than EPSPs that fail to trigger an AP (gray, EPSP$_{fail}$). **Middle**: EPSP rising slopes of EPSP$_{succ}$ (blue) and EPSP$_{fail}$ (gray) show considerable overlap, with EPSP$_{fail}$ having consistently smaller rising slopes than EPSP$_{succ}$. Note the mono-modal, Gaussian distribution of all EPSP inputs (orange), suggesting that both types of events originate from the same source. **Right**: Population data of 62 units: EPSP falling slopes show completely different value ranges (right, p<0.001) which made it possible to clearly separate the two types of events. The respective EPSP rising slopes show considerable overlap (left), but still, the rising slopes of EPSP$_{succ}$ were consistently higher than for EPSP$_{fail}$ (p<0.001). Triangles indicate the respective values of the representative cell on the left. Box plots show medians, interquartile and minimum/maximum values. (**C**) Left: During spontaneous activity, not all EPSPs trigger a postsynaptic AP, gray dots indicate EPSP$_{fail}$, blue dots indicate EPSP$_{succ}$. **Middle**: When stimulated at CF, the discharge rate increases, but the ANF-SBC synapse becomes increasingly unreliable indicated by a high proportion of EPSP$_{fail}$. **Right**: Population data show the considerable variance of failure fraction during spontaneous activity, and a consistent increase in failure fraction during acoustic stimulation. CF = characteristic frequency. Dots indicate values > 1.5 interquartile range.

The following source data is available for figure 1:

**Source data 1.** Rising and falling slope for EPSP$_{fail}$ and EPSP$_{succ}$, and failure fractions during spontaneous activity and acoustic stimulation.

activity, the failure fraction, defined as the proportion of EPSP$_{fail}$ of the ANF input (EPSP$_{succ}$ + EPSP$_{fail}$) amounted to 0.28 [0.11, 0.54] with considerable variability between cells (range: 0.01 to 0.91). Acoustic stimulation at the unit's CF at 50 dB SPL, i.e. within the excitatory response area, increased the failure fraction to 0.49 [0.41, 0.59] rendering the ANF-SBC synapse less reliable during acoustic stimulation ($\Delta$ = 0.18 ± 0.29, p<0.001, paired *t*-test, n = 62, U1 = 0.2, *Figure 1C*).

## Synaptic depression alone fails to account for increased failure rates

The increased incidence of failures during acoustic stimulation has been attributed to the activation of inhibitory inputs (*Kopp-Scheinpflug et al., 2002*; *Kuenzel et al., 2011*, *2015*; *Keine and Rübsamen, 2015*). However, also in vitro experiments need to be considered, which showed strong depression at the ANF-SBC synapse (*Wang and Manis, 2008*; *Yang and Xu-Friedman, 2008*, *2009*; *Wang et al., 2010*) affecting SBC responsiveness for up to tens of milliseconds (*Yang and Xu-Friedman, 2015*). Such depression might also suppress SBC spiking in vivo and result in an increased failure fraction during acoustic stimulation. Still, in vivo the impact of depression was shown to be smaller, since ongoing spontaneous activity – completely absent in slice recordings – seems to keep the synapse in a chronically depressed state (*Hermann et al., 2007*; *Lorteije et al., 2009*; *Yang and Xu-Friedman, 2015*). Also, the in vivo calcium concentration was reported to be lower than in the artificial cerebrospinal fluid usually used in slice studies resulting in lower vesicle release probabilities and thus smaller depression (*Borst, 2010*; *Kuenzel et al., 2011*; *Friauf et al., 2015*).

To determine the cause of altered reliability of synaptic transmission at the ANF-SBC synapse, and to dissect the effect of acoustically evoked inhibition from synaptic depression, we first quantified the dependence of the EPSP rising slope on the preceding spontaneous activity. As indicated above, the rising slopes of $EPSP_{fail}$ and $EPSP_{succ}$ differ, but still show a considerable range of overlap. For each unit, the EPSP rising slopes were pooled for $EPSP_{succ}$ and $EPSP_{fail}$ and binned. Then, the fraction of $EPSP_{succ}$ was calculated for each bin, and a Boltzmann function was fitted to the $EPSP_{succ}$ probability distribution (*Figure 2A* left). The symmetric inflection point of this function indicates the threshold EPSP, i.e. the EPSP slope necessary to trigger an AP with >50% probability. EPSP rising slopes showed strong depression for inter-event intervals (IEI) < 2 ms resulting in high AP failure rates. But, already for IEIs > 5 ms, the preceding activity had only a minor influence on EPSP rising slopes (*Figure 2A*, middle). Averaging the normalized EPSP slopes across cells showed a facilitating effect for IEI between 2 ms and 20 ms (green markers, *Figure 2A* right). The threshold EPSP was increased for IEIs < 2 ms, but not for longer IEIs (black line) and the increase in threshold EPSP resulted in an increased failure fraction for IEIs < 2 ms (orange histogram). While IEIs up to 20 ms resulted in increased EPSP slopes, the effect of IEIs on threshold EPSP and failure fraction was limited to short IEIs < 2 ms. These data are consistent with previous studies, suggesting the presence of short-term facilitation rather than depression of synaptic events. Considering only the last preceding IEI, however, disregards the potential impact of previous medium- and short-term afferent activity. Also, in vitro studies yielded the influence of short-term depression at the ANF-SBC synapse to extend well beyond the last IEI (*Yang and Xu-Friedman, 2015*).

To determine the impact of preceding activity on EPSP strength and AP generation in vivo, the preceding activity of each event was quantified as a weighted sum of all previous events, using an exponentially decaying kernel with a time constant of 60 ms, emphasizing temporally closer events over more distant ones (*Figure 2B*). The analysis yielded only minor influences of preceding activity on EPSP rising slopes on both $EPSP_{fail}$ (gray) and $EPSP_{succ}$ (blue) as shown in a representative cell in *Figure 2C* (left and middle) and also evidenced for the population of recorded units (Spearman's rho $EPSP_{fail}$ = 0.22 ± 0.16 vs. $EPSP_{succ}$ = 0.24 ± 0.11, Δ = 0.02 ± 0.12, p=0.27, paired *t*-test, n = 62, U1 = 0.09, *Figure 2D*, see also *Figure 2—source data 1*). A small but consistent effect, seen in 61/62 recorded units (98%), was a positive correlation (p<0.001) between preceding ANF activity and EPSP rising slopes indicating a facilitating rather than a depressive influence of higher activity levels in vivo.

Postsynaptic spike depression may also contribute to the increase in postsynaptic spike failures ($EPSP_{fail}$). When analyzing the dependence of AP amplitude on preceding SBC spiking activity (exemplary unit shown in *Figure 2C* right) a significant negative correlation was observed in 92% of the cells (57/62) indicating smaller AP amplitudes after periods of higher SBC activity (*Figure 2D*). The representative unit shown in *Figure 2C* (right) shows the respective change in AP amplitude and an inverse effect on the amplitudes of $EPSP_{fail}$, consistent with the facilitating influence on EPSP rising slopes. These results are in agreement with previous reports (*Kuenzel et al., 2011*, *2015*) suggesting that endbulbs are mostly in a close-to-threshold state and show low synaptic depression in vivo.

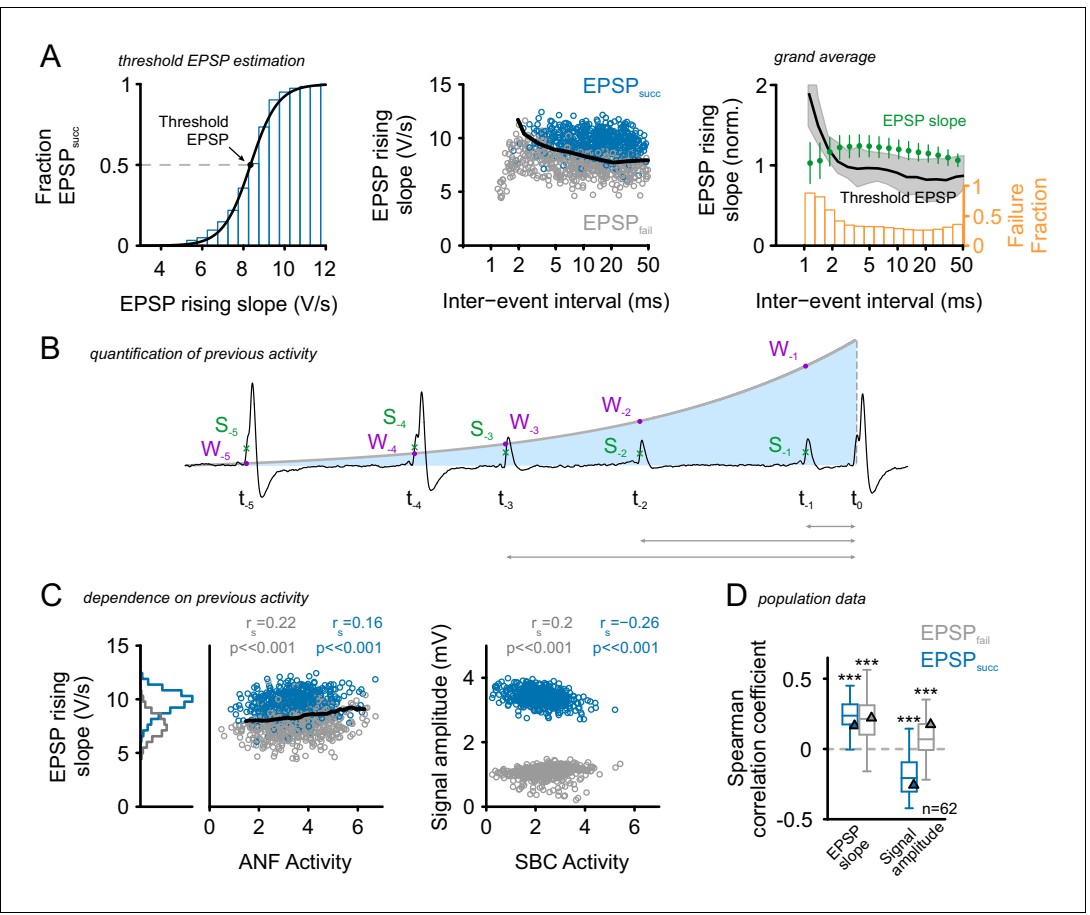

**Figure 2.** Preceding activity has only a minor, facilitating influence on EPSP rising slopes. (**A**) **Left**: Estimation of threshold EPSP for a representative cell: The EPSP rising slopes were binned (0.5 V/s bin size) and the proportion of EPSP$_{succ}$ calculated for each bin. A Boltzmann function was fit to these data. The symmetric inflection point of this function was considered the threshold EPSP and indicates the EPSP rising slope necessary to generate an AP with >50% probability. **Middle**: The inter-event-interval (IEI) between synaptic inputs had only a small influence on the EPSP rising slope, with small IEIs being correlated with moderately increased EPSP rising slopes. A more prominent difference was observed between EPSP$_{fail}$ (gray) which showed consistently smaller rising slopes than EPSP$_{succ}$ (blue) and these differences prevailed over a wide range of IEIs. For IEI < 2 ms the SBCs relative refractoriness renders virtually all EPSPs unsuccessful in triggering a postsynaptic AP. The black line indicates the threshold EPSP. **Right**: Grand average of normalized EPSP slope, threshold EPSP (left ordinate), and failure fraction (right ordinate) in dependence of preceding IEI pooled for EPSP$_{succ}$ and EPSP$_{fail}$ (n = 62 cells). The average EPSP slope (green, left ordinate) showed facilitation for IEIs between 2–20 ms (error bars indicate standard deviation). The median threshold EPSP (black line, left ordinate) was elevated only for IEIs < 2 ms and well below average EPSP size for larger IEIs (shaded area indicates first and third quartile). The elevated threshold EPSP resulted in an increased failure fraction for IEIs < 2 ms, while for longer IEIs the reliability of AP generation seemed not to be affected (orange, right ordinate). (**B**) Consideration of a wider time span of preceding activity: Sketch of the quantification of preceding activity by exponentially weighting [W] all preceding EPSP rising slopes [S] (ANF activity) or AP amplitudes (SBC activity) depending on the distance to the event under investigation. (**C**) **Left**: EPSP rising slopes for the representative cell showed only minor dependence on previous ANF activity levels. Note that the EPSP$_{fail}$ (gray) showed consistently lower EPSP rising slopes (histogram on the left); still, the EPSPs slopes tend to increase during periods of high activity. The threshold EPSP (black line) increased as a function of ANF activity. Threshold EPSP was calculated for different levels of ANF activity (bin size = 0.5). **Right**: signal amplitudes of EPSP$_{fail}$ (gray) and APs (blue) as a function of preceding SBC activity showed decreasing AP amplitudes but increasing EPSP amplitudes. (**D**) Population data for 62 units (n = 62): While EPSP slopes tend to be elevated after periods of high activity (left), AP amplitudes showed a negative correlation with preceding SBC activity (p<0.001, one-sample t-test against zero). Triangles indicate the data of the representative cell. Organization of the graph as described above. .

*Figure 2 continued on next page*

*Figure 2 continued*

The following source data is available for figure 2:

**Source data 1.** Correlation between EPSP slopes and signal amplitudes on preceding ANF activity.

These results suggest that the increased failure fraction during acoustic stimulation in vivo is not explainable by endbulb depression evoked by high firing rates, highlighting the role of acoustically evoked inhibition on the input-output relationship at the ANF-SBC synapse.

## Broadband on-CF inhibition shapes SBC tuning

Frequency response areas (FRA) of SBCs show prominent inhibitory sidebands and reduced firing activity in the excitatory field compared to the ANF input (*Kopp-Scheinpflug et al., 2002*; *Kuenzel et al., 2011*; *Keine and Rübsamen, 2015*). Also, about half of the SBCs show pronounced non-monotonic rate-level functions pointing to an impact of inhibition (*Kopp-Scheinpflug et al., 2002*; *Keine and Rübsamen, 2015*; *Kuenzel et al., 2015*) which has been further classified as 'on-CF inhibition' and 'broadband inhibition' (*Winter and Palmer, 1990*; *Caspary et al., 1994*; *Kopp-Scheinpflug et al., 2002*). In vitro and modeling studies showed that glycinergic inhibition can elevate the threshold EPSP for AP initiation (*Xie and Manis, 2013*; *Kuenzel et al., 2015*). Thus the threshold EPSP can serve as a suitable indicator for the activation of inhibitory inputs.

In the present loose-patch recordings, elevation in threshold EPSP (*Figure 3Aii/iii*) was observed throughout the FRAs (*Figure 3Ai/iii*) accompanied by an increase in failure fraction (*Figure 3Bii*). The frequency profile of threshold EPSP elevation closely matched the one of increased failure fraction (*Figure 3Aii/Bii*, Spearman correlation $r_s$ = 0.7 [0.4, 0.76], p<0.001, Wilcoxon signed rank test, n = 62, U1 = 0.98, population data not shown). The FRA of threshold EPSP elevation was used to quantify the inhibitory influence, which was then compared to the SBC's excitatory FRA. Both FRAs had similar CFs, defined as the stimulus frequency at which the lowest sound intensity resulted in a significant increase in ANF firing rate (excitatory) or threshold EPSP (inhibitory) (2.2 ± 0.6 kHz vs. 2.2 ± 0.9 kHz, respectively, Δ = 0.03 ± 0.83 kHz, p=0.77, paired *t*-test, n = 62, U1 = 0.05, *Figure 3Ci*), but inhibitory FRAs exhibited higher thresholds (excitatory = 4.8 ± 6.1 dB SPL vs. inhibitory = 19.8 ± 16.6 dB SPL, Δ = 15 ± 15.5 dB SPL, p<0.001, paired *t*-test, n = 62, U1 = 0.2, *Figure 3Cii*, see also *Figure 3—source data 1*).

The width of inhibitory and excitatory FRA was determined by calculating $Q_{10}$ and $Q_{40}$ values. Both measures were smaller for the inhibitory FRA compared to the excitatory FRA, i.e. inhibition showed reduced frequency selectivity compared to excitation ($Q_{10}$: excitatory = 2.5 ± 0.6 vs. inhibitory = 1.9 ± 1.1, Δ = 0.5 ± 1.2, p<0.01; $Q_{40}$: excitatory = 0.96 ± 0.1 vs. inhibitory = 0.6 ± 0.3, Δ = 0.4 ± 0.3, p<0.001, two-way RM ANOVA, Bonferroni-adjusted, n = 62, $\eta^2$ = 0.06, *Figure 3Ciii*).

Q-values provide information about the sharpness of tuning, but not about the actual shape of the FRA. The tuning shape was evaluated using the asymmetry index (AI, see Materials and methods), with values of 0 indicating symmetric, <0 for low-frequency extended and >0 for high frequency extended tuning curves. While excitatory FRAs showed distinct low-frequency tails, typical for ANF, inhibitory FRAs were mostly symmetrically arranged around CF, partly covering high-frequency ranges above the excitatory response area (AI excitation = –0.96 ± 0.42 vs AI inhibition = –0.26 ± 0.88, Δ = 0.7 ± 0.97, p<0.001, paired *t*-test, n = 62, U1 = 0.21, *Figure 3Civ*).

The rate-level function (RLF) of the SBC output was markedly flatter and thus less variable with respect to level than the rate-level function of the excitatory ANF input. The gain of the neuronal response across stimulus level was quantified as rate level gain (RLG), defined as $\mathrm{RLG} = log_{10}\left(\frac{FR_{max}-FR_{min}}{FR_{spont}}\right)$, with $FR_{max}$ and $FR_{min}$ being the maximal and minimal firing rate in the RLF and $FR_{spont}$, the spontaneous firing rate in the absence of acoustic stimulation (see also supplementary Matlab code). This way, overall changes in firing rates are taken into account (e.g. due to spontaneous failures). The output's rate level function had a gain of 1 ± 0.4 which was significantly less than the input's (1.4 ± 0.4, Δ = 0.35 ± 0.3, p<0.001, paired *t*-test, n = 62, U1 = 0.1, *Figure 3D*).

Taken together, these data demonstrate that inhibition in SBC is co-tuned with excitation and shows a broader and more symmetric frequency profile, which results in flatter rate-level functions and high-frequency inhibitory sidebands at the fringes of the tuning curve (frequently observed in

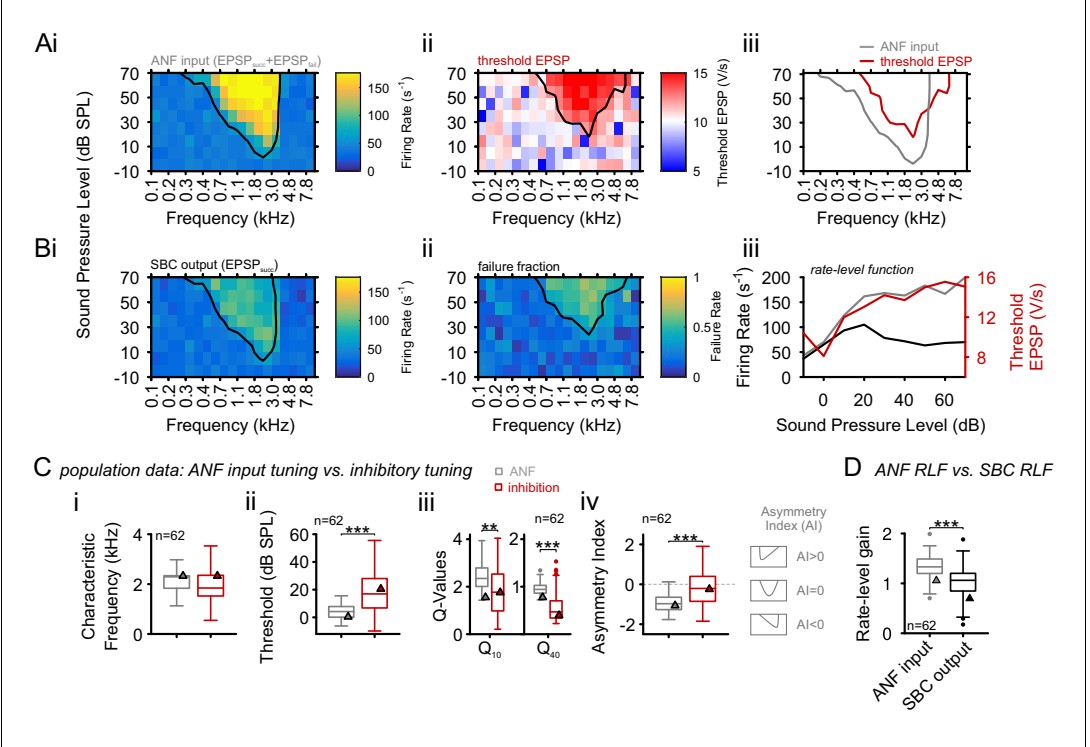

**Figure 3.** Inhibition at SBC is co-tuned with excitation and broadband, not off-CF and narrowband. (A) **i**: Representative frequency response area (FRA) of the excitatory ANF input ($EPSP_{fail}$ and $EPSP_{succ}$) characterized by a well-defined CF, the typical steep high-frequency flank, the formation of a low-frequency tail, and the absence of frequency-intensity domains of inhibition. **ii**: The same recording showed elevated threshold EPSPs throughout most of the excitatory response area and extending up to two octaves above CF. The frequency, where the lowest relative intensity caused elevated threshold EPSP, matched the units CF. **iii**: For the same unit, comparison of excitatory (ANF, gray) response area and frequency-intensity domain of inhibition (threshold EPSP elevation, red). The inhibitory domain was symmetrically arranged around the unit's CF. (B) **i**: FRA of the SBC output ($EPSP_{succ}$) shows a considerable reduction in firing activity compared to the ANF input. **ii**: Failure fraction, i.e. the proportion of $EPSP_{fail}$. The increase in failure fraction is most prominent around the units CF. Note the similarity of the frequency-intensity domains of EPSP threshold increase and the respective domains with increased $EPSP_{fail}$ in Aii. **iii**: Rate-level functions of ANF input (gray line, left ordinate) and SBC output (solid black line, left ordinate) compared to threshold EPSP (red, solid line, right ordinate). Increasing sound pressure levels result in a monotonic increase in ANF firing and correspondingly the threshold EPSP shows a monotonic increase. The SBC output is maximal at 20 dB SPL and declines towards higher stimulus intensities. (C) Population data: comparison of excitatory (ANF, gray) and inhibitory (threshold EPSP, red) FRA indicates (**i**) on-CF inhibition although (**ii**) with higher thresholds (p<0.001, paired t-test), which is (**iii**) broadly tuned (Q10: p<0.01, Q40: p<0.001, two-way RM ANOVA), and (**iv**) shows a more symmetric tuning (p<0.001, paired t-test; the schematic drawing on the right indicates FRA shapes for different asymmetry indices). (D) Finally, the rate-level functions were shallower and showed a reduced gain in firing rate in the output compared to the input. SBC = spherical bushy cell, CF = characteristic frequency, EPSP = excitatory postsynaptic potential, ANF = auditory nerve fibers, FRA = frequency response area.

The following source data is available for figure 3:

**Source data 1.** Tuning properties of excitatory and inhibitory inputs onto SBCs.

SBC output activity). These findings are consistent with previous reports (*Caspary et al., 1994*; *Kopp-Scheinpflug et al., 2002*; *Kuenzel et al., 2011*). The lower rate-level dependence suggests a gain-normalization function of inhibition, discussed in detail below.

## Acoustically evoked inhibition elevates threshold for AP generation

As shown above, the tuning of the inhibitory input on SBCs largely matches the ANF excitation. In vitro studies and modeling suggested inhibition to prevent AP generation in SBC by elevating the threshold EPSP (*Xie and Manis, 2013*; *Kuenzel et al., 2015*). Slice studies reported a predominately glycinergic inhibition with a smaller GABAergic contribution (*Nerlich et al., 2014a*, *2014b*), and in vivo studies showed an effective, dose-dependent block of SBC spiking by iontophoretic application of glycine (*Keine and Rübsamen, 2015*). Consequently, we tested if the activation of glycinergic

inputs can directly cause the observed elevation of threshold EPSP. Glycine was applied iontophoretically, mimicking the putative role of glycinergic inhibition, while monitoring the SBC's spontaneous activity. Indeed, glycine caused an increase in the number of EPSPs that failed to trigger APs, and this specific effect could be blocked by simultaneous application of the glycine receptor antagonist strychnine (*Figure 4A*). The iontophoretic current for glycine application was adjusted to cause an increase in the spontaneous failure fraction from $0.3 \pm 0.17$ to $0.64 \pm 0.18$ ($\Delta = 0.34 \pm 0.12$, p<0.001, paired *t*-test, n = 11, U1 = 0.5) to match the range observed under acoustic stimulation. This increase in failure fraction was accompanied by an elevation in threshold EPSP (threshold EPSP spont = $6.1 \pm 2.2$ V/s vs threshold EPSP glycine = $8.4 \pm 1.8$ V/s, $\Delta = 2.3 \pm 0.9$ V/s, p<0.001, paired *t*-test, n = 11, U1 = 0.32, *Figure 4B*, see also *Figure 4—source data 1*). The application of the carrier alone had neither an effect on threshold EPSPs (threshold EPSP spont = $6.8 \pm 1.6$ V/s vs. threshold EPSP carrier = $6.8 \pm 1.7$, $\Delta = 0 \pm 0.3$, p=0.89, paired *t*-test, n = 9, U1 = 0.1, data not shown) nor on failure fraction (failure fraction spont = $0.29 \pm 0.17$ vs. failure fraction carrier = $0.27 \pm 0.16$, $\Delta = 0.03 \pm 0.06$, p=0.79, paired *t*-test, n = 9, U1 = 0.11, data not shown).

Next, the contribution of inhibition-mediated threshold EPSP elevation on spike failures during acoustic stimulation was tested. The specific glycine receptor antagonist strychnine was iontophoretically applied to block the acoustically evoked glycinergic inhibition. The effectiveness of the glycine block was tested before sound stimulation by simultaneously applying glycine and strychnine, with the application current for strychnine adjusted to block the effect of iontophoretically applied glycine. The ANF input firing rates were not influenced by the block of inhibition (control = $282 \pm 48$ Hz vs. strychnine = $283 \pm 50$ Hz, $\Delta = 1.2 \pm 26.7$ Hz, p=0.88, paired *t*-test, n = 11, U1 = 0.18, *Figure 4C*). The SBC output rates in the excitatory field, however, were substantially increased under glycine block (control = $122 \pm 49$ Hz vs. strychnine = $192 \pm 39$ Hz, $\Delta = 70 \pm 47.9$ Hz, p<0.001, paired *t*-test, n = 11, U1 = 0.5, *Figure 4D*). We next tested if the block of glycinergic inhibition differentially affects the threshold EPSP during spontaneous activity and during acoustic stimulation. When glycinergic inhibition was blocked, the threshold EPSP was only affected during acoustic stimulation, but not during spontaneous activity (interaction drug × stimulus condition, p<0.01, $\eta^2 = 0.13$, two-way RM ANOVA, Greenhouse-Geisser corrected, n = 11, *Figure 4E*). Acoustic stimulation at CF under control condition resulted in a significant threshold EPSP elevation (threshold EPSP spont = $5.4 \pm 1.6$ V/s vs. stim = $8.8 \pm 3.1$ V/s, $\Delta = 3.5 \pm 2.9$ V/s, p<0.01, two-way RM ANOVA, Bonferroni-adjusted, n = 11, U1 = 0.41) and this shift was absent when the inhibition was blocked (threshold EPSP spont = $5.9 \pm 2$ V/s vs. stim = $5.8 \pm 1.8$ V/s, $\Delta = 0.1 \pm 1.3$ V/s, p=0.82, two-way RM ANOVA, Bonferroni-adjusted, n = 11, U1 = 0.09, *Figure 4D*). The effects observed under acoustic stimulation were very different from the respective manipulations performed during spontaneous activity. In the absence of acoustic stimulation, the block of glycinergic inhibition had no effect on output rates (control = $51 \pm 26$ Hz vs. strychnine = $49 \pm 20$ Hz, $\Delta = 1.6 \pm 18.5$ Hz, p=0.79, two-way RM ANOVA, Bonferroni-adjusted, n = 11, U1 = 0.14, data not shown), and threshold EPSP (control = $5.5 \pm 2.5$ V/s vs strychnine = $5.6 \pm 2.6$ V/s, $\Delta = 0.03 \pm 0.16$ V/s, p=0.53, two-way RM ANOVA, Bonferroni-adjusted, n = 11, U1 = 0.09, *Figure 4Eii*). Similar to acoustic stimulation, the input rates were not altered during inhibition block (control = $85 \pm 26$ Hz vs. strychnine = $86 \pm 22$ Hz, $\Delta = 0.4 \pm 16.6$ Hz, p=0.94, two-way RM ANOVA, Bonferroni-adjusted, n = 11, U1 = 0.09, data not shown).

These data suggest a major role of glycinergic inhibition in acoustically evoked signal processing, but a negligible impact during spontaneous activity. Taken together, the data confirms previous reports of broadly tuned, predominantly glycinergic inhibition (*Kopp-Scheinpflug et al., 2002*; *Kuenzel et al., 2011*), which decreases and potentially normalizes SBC output firing across different stimulus conditions by an increase in threshold EPSP for spike generation.

## Temporal precision improves from ANF to SBC during amplitude and frequency-modulated tones

The results above suggest that acoustically evoked inhibition can considerably influence SBC spiking by increasing the threshold for AP generation. Previous studies directly comparing the ANF input and SBC output showed an increase in temporal precision which has been attributed to the impact of inhibition (*Dehmel et al., 2010*; *Kuenzel et al., 2011*; *Keine and Rübsamen, 2015*). These studies focused on the responses to static pure-tone stimulation leaving the question for a potential

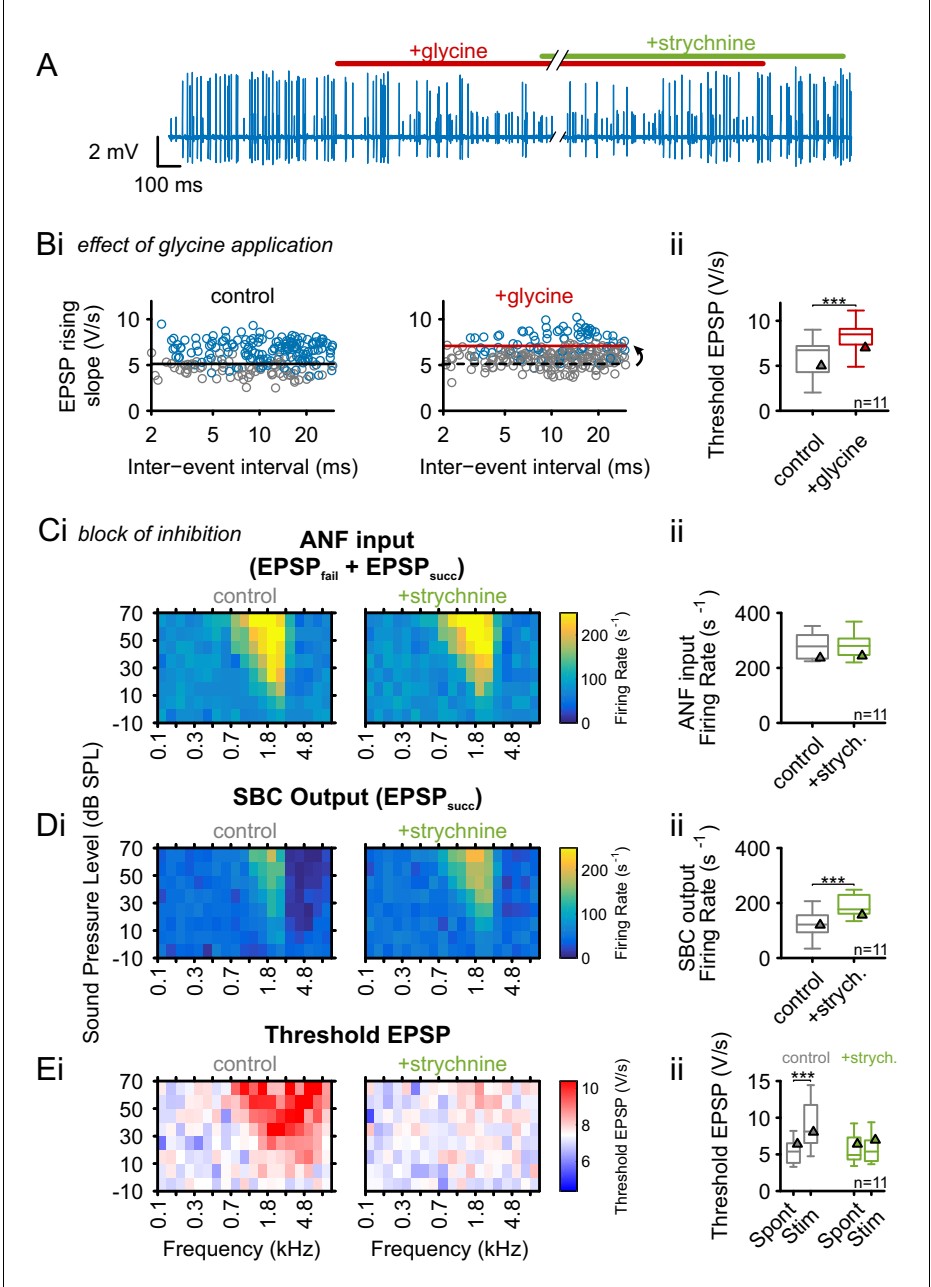

**Figure 4.** Glycinergic inhibition elevates threshold EPSP and becomes activated during acoustic stimulation. (A) Representative recording of spontaneous activity with iontophoretically applied glycine to block SBC spiking (red bar). This effect is suspended by strychnine application (green bar). (B) (Bi) Left: During spontaneous activity, small EPSPs fail to generate APs (gray = EPSP$_{fail}$, blue = EPSP$_{succ}$, black line = threshold EPSP). Right: Iontophoretic application of glycine elevates the threshold EPSP (solid red line) for spike generation resulting in an increased failure fraction (dashed black line shows threshold EPSP from control condition). (Bii): Population data for 11 units showing the effect of glycinergic inhibition on the increase of threshold EPSP. (C) and (D): Acoustically evoked FRAs while blocking glycinergic inhibition. (Ci) No effect on input FRA was observed when inhibition was blocked. (Cii) Population data confirming the lack of glycine effect on the input activity. (Di) SBC output FRA shows increased firing rates during the blockade of glycinergic inhibition. Note the absence of the inhibitory sideband after inhibition block. (Dii) Population data show a considerable increase in SBC firing after block of inhibition (p<0.001, paired t-test). (Ei) Left: Under control condition, t threshold EPSP is elevated during acoustic stimulation, indicating the presence of acoustically evoked inhibition. Right: This threshold elevation is absent when the glycinergic inhibition is blocked. (Eii) Population data showing the threshold EPSP during spontaneous activity and acoustic stimulation at the units' CF for control condition (gray, p<0.001, two-way RM ANOVA) and under

*Figure 4 continued on next page*

*Figure 4 continued*

inhibition block (green). Note the absence of threshold EPSP elevation during acoustic stimulation under the inhibition block. Blocking glycinergic inhibition had no effect on threshold EPSP during spontaneous activity. Triangles in **Bii–Eii** denote representative cells from **Bi–Ei**.

The following source data is available for figure 4:

**Source data 1.** Iontophoretic application of glycine and strychnine.

influence of acoustically evoked inhibition on signal transmission in a more complex, i.e. a more naturalistic acoustic environment unaddressed.

Considering this issue, we first tested the responses of SBCs to sinusoidal amplitude-modulated (SAM) and frequency-modulated (SFM) acoustic stimuli. SAM stimuli were presented at the respective units' CF 30 dB above the excitatory threshold with modulation frequencies between 50 Hz and 400 Hz (modulation depth = 100%, *Figure 5*). The discharge activity of the units showed different degrees of modulation congruent with the SAM for both the ANF input and the SBC output (*Figure 5B,C*). The AP failure fraction increased from $0.27 \pm 0.22$ in the absence of acoustic stimulation to $0.43 \pm 0.18$ during SAM stimulation ($\Delta = 0.16 \pm 0.18$, p<0.01, two-way RM ANOVA, Greenhouse-Geisser corrected, n = 14, $\eta^2 = 0.11$, data not shown) and was independent of modulation frequency (factor frequency: p=0.19, $\eta^2 < 0.01$; interaction stimulus type $\times$ frequency: p=0.27, two-way RM ANOVA, Greenhouse-Geisser corrected, $\eta^2 < 0.01$, n = 14). The temporal precision of ANF input and SBC output to SAM stimulation was estimated by calculating the vector strength (VS) at different modulation frequencies. The SBC output exhibited consistently higher VS compared to its ANF input ($\Delta = 0.06 \pm 0.04$, p<0.001, two-way RM ANOVA, Greenhouse-Geisser corrected, n = 14, $\eta^2 = 0.09$, *Figure 5D*, see also *Figure 5—source data 1*), and decreased for modulation frequencies above 200 Hz (factor frequency: p<0.001, $\eta^2 = 0.09$; interaction signal type $\times$ frequency: p<0.05, $\eta^2 < 0.01$, two-way RM ANOVA, Greenhouse-Geisser corrected, n = 14). To estimate the degree of modulation of the neural response, the modulation depth was estimated by calculating the standard deviation of the first cycle of the normalized cross-correlation function. Modulation depth was considerably higher at the SBC output ($\Delta = 0.04 \pm 0.04$, p<0.001, $\eta^2 = 0.09$) and decreased with modulation frequency (factor frequency: p<0.001, $\eta^2 = 0.08$; interaction signal type $\times$ frequency: p=0.12, $\eta^2 < 0.01$, two-way RM ANOVA, Greenhouse-Geisser corrected, n = 14).

The neuronal response to a given stimulus can vary between identical stimulus presentations. This trial-to-trial variability was quantified by calculating the within-cell, across-trial crosscorrelations separately for the ANF input and SBC output. The peak height of the crosscorrelation was termed reproducibility (*Joris et al., 2006*). It provides a measure of how repeatable the neural response is across trials, given identical stimulus presentations. If the reproducible features of the response encode stimulus properties, e.g. certain salient events, then an increased reproducibility corresponds to more trustable encoding of stimulus information across trials.

The analysis revealed higher reproducibility in the SBC output compared to the ANF input ($\Delta = 0.4 \pm 0.25$, p<0.001, $\eta^2 = 0.09$) and also showed a systematic decrease with increasing modulation frequency (factor frequency: p<0.001, $\eta^2 = 0.04$; interaction signal type $\times$ frequency: p<0.01, $\eta^2 < 0.01$, two-way RM ANOVA, Greenhouse-Geisser corrected, n = 14). To obtain a better understanding of how precisely the neuronal response reproduces the stimulus envelopes, the delay-adjusted period histograms were correlated to the stimulus envelope resulting in a $Corr_{Norm}$ between 0.82 and 1 (see Materials and methods for explanation). The analysis revealed higher $Corr_{Norm}$ for the SBC output compared to the ANF input ($\Delta = 0.01 \pm 0.01$, p<0.05, $\eta^2 = 0.02$), which for both signal types increased with modulation frequency (factor frequency: p<0.05, $\eta^2 = 0.08$; interaction signal type $\times$ frequency: p=0.19, $\eta^2 < 0.01$, two-way RM ANOVA, Greenhouse-Geisser corrected, n = 14). These analyses show that the increase in temporal precision observed during pure-tone stimulation is maintained during amplitude-modulated sounds across a wide range of modulation frequencies.

In a next step, the modulation of unit discharges to periodic frequency modulations (SFM) was explored. For that purpose, the stimulus intensity was fixed at 30–40 dB above the unit's threshold and the stimulus frequency modulated between one octave below and two octaves above the unit's

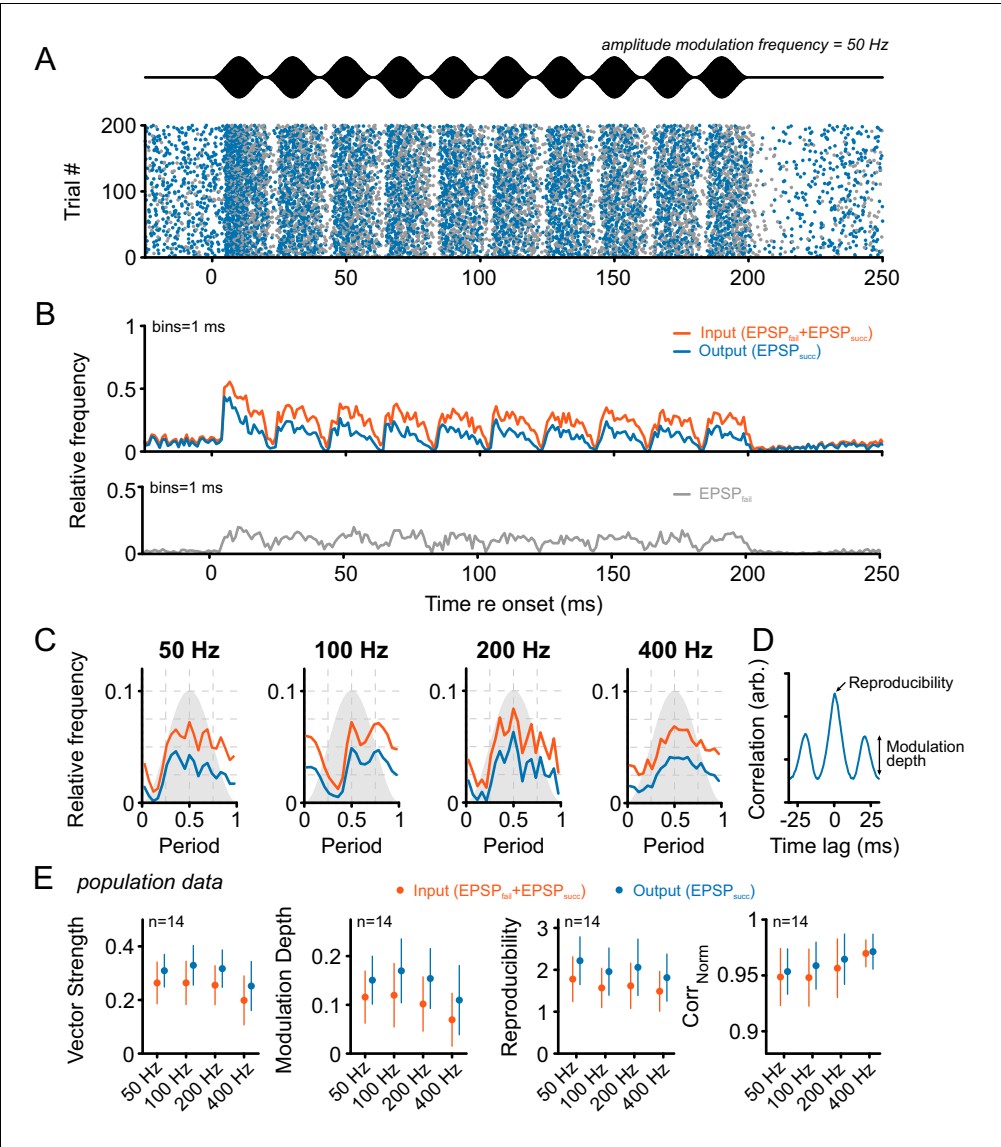

**Figure 5.** Tone bursts with sinusoidal amplitude modulations (SAM) of different modulation frequencies were used to investigate the input-output function under the condition of dynamically altered amplitude profiles. Overall, SAM testing revealed higher temporal precision and reproducibility from ANF input to SBC output. (**A**) The upper panel (black) shows the stimulus and the lower panel the dot-raster plot of the discharges of a representative SBC to 200 stimulus presentations with a differentiation between EPSP$_{succ}$ (blue) and EPSP$_{fail}$ (gray). (**B**) Histogram of the discharge activity shown in A. **Upper panel**: blue = EPSP$_{succ}$, orange = ANF input, i.e. EPSP$_{succ}$+EPSP$_{fail}$). **Lower panel**: The EPSP$_{fail}$ is also locked to the SAM, following the ANF input dynamics. (**C**) Period histograms of ANF input (orange) and SBC output (blue) to increasing modulation frequencies. For comparison, all histograms are centered to the maximum of the ANF input. Gray background indicates the stimulus modulation. (**D**) Trial-to-trial reproducibility and modulation depth were calculated from the cross-correlation between trials. Reproducibility was defined as the peak of the normalized cross correlation and modulation depths as the standard deviation of the first cycle. (**E**) Population data for 14 SBCs. Different measures of temporal precision and trial-to-trial reproducibility all revealed higher accuracy for the SBC output compared to its ANF input: The SBC output showed consistently higher vector strength (left, $p<0.001$, two-way RM ANOVA), increased modulation depth (middle left, $p<0.001$, two-way RM ANOVA), higher reproducibility (middle right, $p<0.01$, two-way RM ANOVA) and higher representation of the stimulus envelope (right, $p<0.05$, two-way RM ANOVA) throughout all modulation frequencies. Markers indicate mean ± standard deviation.

The following source data is available for figure 5:

**Source data 1.** Metrics of temporal precision and reproducibility during SAM stimulation.

CF. This frequency range covers the whole excitatory area as well as the inhibitory sideband. Similar to the SAM stimulation, the SFM resulted in prominent modulations of the units' firing rates (*Figure 6A*) and increased failure fractions (*Figure 6B*) (spont = 0.34 ± 0.25 vs. stim = 0.6 ± 0.13, Δ = 0.26 0.23, p<0.001, $\eta^2$ = 0.29, data not shown). In contrast to SAM stimulation, SFM led to increased failure rates at higher modulation frequencies (e.g. 0.52 ± 0.13 at 20 Hz vs. 0.65 ± 0.12 at 400 Hz modulation frequency, p<0.001, two-way RM ANOVA, Greenhouse-Geisser corrected, $\eta^2$ = 0.02, n = 19, data not shown). For SFM stimulation – same as for SAM - the SBC output showed higher VS compared to their ANF input (Δ = 0.14 ± 0.09, p<0.001, $\eta^2$ = 0.21, see also *Figure 6— source data 1*) (*Figure 6D* left) (factor frequency: p<0.001, $\eta^2$ = 0.48; interaction signal type × frequency: p<0.001, $\eta^2$ = 0.03, two-way RM ANOVA, Greenhouse-Geisser corrected, n = 19). Still, overall the VS of ANF input and SBC output decreased with increasing modulation frequency (*Figure 6D* left). Notably, the VS of the SBC output deteriorated to a lesser degree than the ANF input. At modulation frequencies of 20 Hz, the output VS was not significantly different between ANF input and SBC output (ANF input = 0.61 ± 0.06 vs. SBC output = 0.63 ± 0.1, Δ = 0.04 ± 0.07, p=0.17, two-way RM ANOVA, Bonferroni-adjusted, n = 19, U1 = 0.16). For modulation frequencies of 400 Hz, however, the VS of the SBC output was considerably higher than the ANF input (ANF input = 0.25 ± 0.08 vs. SBC output 0.4 ± 0.09, Δ = 0.15 ± 0.06, p<0.001, two-way RM ANOVA, Bonferroni-adjusted, n = 19, U1 = 0.5). It has to be considered that the interpretation of VS values is difficult when the period histogram of the neuronal response shows multiple peaks (*Figure 6C*). We, therefore, used a set of additional measures to describe the neuronal response to SFM stimuli when comparing ANF input and SBC output. The modulation depth was considerably higher for the SBC output than the ANF input (*Figure 6D* midleft; Δ = 0.24 ± 0.16, p<0.001, $\eta^2$ = 0.28) and strongly depended on the modulation frequency (factor frequency: p<0.001, $\eta^2$ = 0.37; interaction signal type × frequency: p<0.001, $\eta^2$ = 0.03, two-way RM ANOVA, Greenhouse-Geisser corrected, n = 19). The same holds for signal reproducibility (*Figure 6D* midright; Δ = 1.9 ± 1.2, p<0.001, $\eta^2$ = 0.38) which also showed prominent frequency dependency (factor frequency: p<0.001, $\eta^2$ = 0.26; interaction signal type × frequency: p<0.001, $\eta^2$ = 0.02, two-way RM ANOVA, Greenhouse-Geisser corrected, n = 19). Unlike the previous measures, the normalized correlation between SFM stimulus envelope and neural response revealed a lower reproducibility for the SBC output compared to the ANF input (*Figure 6D* right; Δ = 0.02 ± 0.03, p<0.001, $\eta^2$ = 0.06), and the difference also holds with respect to the effect of modulation frequency (factor frequency: p<0.001, $\eta^2$ = 0.43; interaction signal type × frequency: p<0.001, $\eta^2$ = 0.03, two-way RM ANOVA, Greenhouse-Geisser corrected, n = 19). Overall, these data suggest that – same as for the SAM stimuli – the SBC output shows temporally increased precision and higher response reproducibility during SFM stimulation.

## Spectrotemporal input-output comparison indicates broad, co-tuned, long-lasting inhibition

Above, we demonstrated an improvement in temporal precision and reproducibility in response to SAM and SFM acoustic stimuli. In natural environments, however, the auditory system has to cope with simultaneous dynamic changes in both frequency and amplitude embedded in ambient background noise. To mimic such conditions, while preserving the possibility for a quantifying data analysis, dynamic acoustic stimuli composed of gamma-tones randomly placed in the spectrogram were used (*Figure 7A* top, randomized gamma-tone sequence, RGS, see Materials and methods for details). The SBC activity can then be characterized using spectrotemporal receptive fields (STRFs). In the present context, STRFs can also be used to quantify the spectrotemporal transformation of response properties across the ANF-SBC synapse, since the respective analysis can be performed for both the ANF input and SBC output. SBC activity was recorded while presenting 20–30 repetitions of identically structured RGS sequences of 30 s duration each. In *Figure 7A*, the upper panel shows a 200 ms-section of an RGS stimulus used for stimulation of an SBC with a CF of 2 kHz; the middle panel depicts the spike raster plot differentiating between $EPSP_{succ}$ and $EPSP_{fail}$.

The neural response to gamma-tones of both the ANF input and SBC output were temporally structured (*Figure 7A* second and third panel). Failures of signal transmission (*Figure 7A* gray dots in second panel and histogram in fourth panel) were found to be increased following sequences of activation, suggesting a long-lasting action of inhibition (e.g. *Figure 7A* fourth panel, where $EPSP_{fail}$ shows a considerable increase in the responses to the second of the first two peaks).

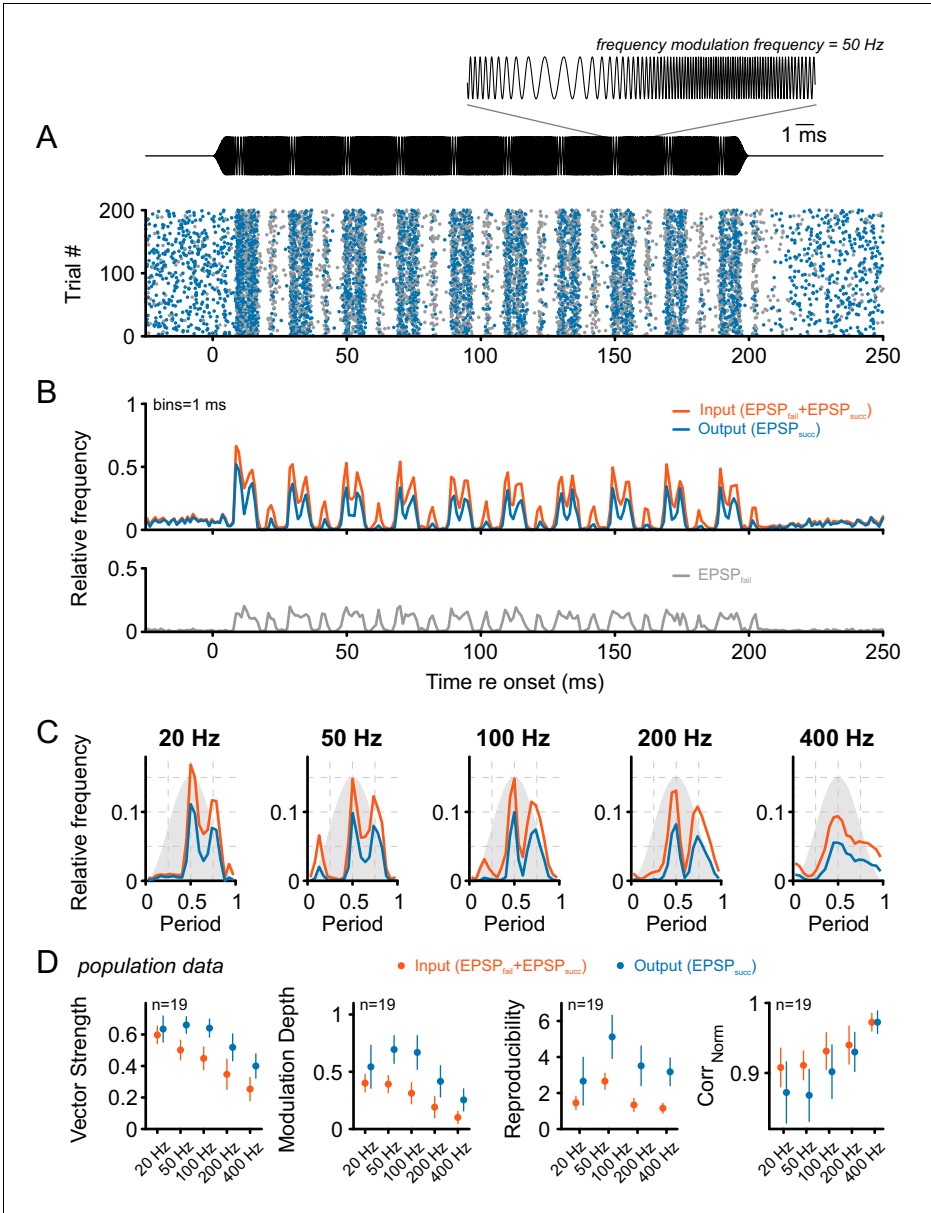

**Figure 6.** Tone bursts with sinusoidal frequency modulations (SFM) of different modulation frequencies were used to investigate the input-output function under the condition of dynamically altered frequency profiles. Overall, SFM testing revealed improved temporal precision and across-trial reproducibility across the ANF-SBC synapse. (A) The upper panel (black) shows the SFM stimulus with a detail enlargement visualizing the dynamic frequency modulation. The dot-raster plot (lower panel) shows the activity of a representative SBC (CF = 1.8 kHz) to 200 stimulus repetitions with a differentiation between EPSP$_{succ}$ (blue) and EPSP$_{fail}$ (gray). (B) Histogram of the discharge activity shown in A. **Upper panel**: blue = EPSP$_{succ}$, gray = EPSP$_{fail}$, orange = ANF input, i.e. EPSP$_{succ}$+EPSP$_{fail}$. **Lower panel**: The EPSP$_{fail}$ is also locked to the SFM but showed reduced fine structure compared to the ANF input. (C) Period histograms for the same cell as in A and B at different modulation frequencies (orange = ANF input, blue = SBC output). Design of the graph is identical to *Figure 5C*. Note the multiple peaks of the response in the period histogram. (D) Population data for 19 cells: Across all frequencies tested, the SBC output shows increased vector strength (left; p<0.001, two-way RM ANOVA), higher modulation depths (mid left; p<0.001, two-way RM ANOVA), and better across-trial reproducibility (mid right; p<0.001, two-way RM ANOVA) compared to its ANF input. The stimulus reproduction (Corr$_{Norm}$) was consistently lower at the SBC level (right; p<0.001, two-way RM ANOVA). Markers indicate mean ± standard deviation.

The following source data is available for figure 6:

*Figure 6 continued on next page*

*Figure 6 continued*

**Source data 1.** Metrics of temporal precision and reproducibility during SFM stimulation.

Separate STRFs were computed for the ANF input (*Figure 7B* middle left) and the SBC output (*Figure 7B* middle right). As expected, both STRFs showed common features, e.g. frequency domain of excitation above and below the unit's CF (in the present example 2 kHz, estimated from single tone tunings) and also the response latency (here 2.5 ms). Importantly, the reduction in the responses establishing a high-frequency sideband was already present in the ANF input to the SBCs and did not become more pronounced in SBC output. Since the ANF activity is not affected by acoustically evoked inhibition, the respective frequency-specific reduction observed in the ANF input to the SBCs likely reflects mechanical interactions in the cochlea, previously described as two-tone suppression (*Engebretson and Eldredge, 1968*; *Sachs and Kiang, 1968*; *Sellick and Russell, 1979*).

To evaluate the signal processing at the ANF-SBC junction, the two STRFs were subtracted from each other after normalizing each by its standard deviation (to compensate for overall firing rate differences, see Materials and methods for details; *Figure 7B* bottom). This normalization allows a quantification of changes in the tuning shape. The increase in $EPSP_{fail}$ in the STRF of the SBC output manifests itself as a broad field of negativity in the difference-STRF around CF extending up to ~10 ms after the onset of the effective signal components around 2 kHz. The respective differences between ANF input and SBC output were quantified in all recorded SBCs (n = 34) separately for the positive (red) and negative (blue) regions in the STRF (corresponding to influential spectrotemporal locations in the stimulus prior to the response). Summing all the positive regions revealed a significant reduction from ANFs to SBCs (ANF = 0.38 [0.37, 0.41] vs. SBC = 0.33 [0.31, 0.36], Δ = 0.05 [0.03, 0.07], p<0.001, Wilcoxon signed rank test, n = 34, U1 = 0.16, *Figure 7C*, see also *Figure 7— source data 1*). Similarly, the summed negative region in the SBC output was significantly larger in magnitude than the ANF input (ANF = 0.18 [0.15, 0.23] vs. SBC = 0.25 [0.18, 0.29], Δ = 0.04 [0.02, 0.07], p<0.001, Wilcoxon signed rank test, n = 34, U1 = 0.1, *Figure 7D*). Together, this suggests an inhibitory influence acting broadly with respect to the neuron's tuning.

We further quantified changes in the shape of the main excitatory peak. The spectral tuning, measured as half-width of the excitatory region, was reduced at the SBC output compared to ANF input, suggesting a spectrally sharper tuning at the SBC output (ANF = 1.2 [0.9, 1.4] octaves vs. SBC = 1 [0.8, 1.2] octaves, Δ = 0.1 [0, 0.2], p<0.001, Wilcoxon signed rank test, n = 34, U1 = 0.01, *Figure 7E*). Temporal precision, measured correspondingly as the half-width of the excitatory region, was somewhat higher for the SBC output, but did not reach statistical significance (ANF = 2.2 [1.8, 3.4] ms vs. SBC = 2.0 [1.8, 3.1] ms, Δ = 0.08 [-0.1, 0.18], p=0.16, Wilcoxon signed rank test, n = 34, U1 = 0.03, *Figure 7F*).

The overall shape of the difference-STRF of all units was studied by aligning all STRFs to the peak excitation and averaging them (*Figure 7G*). As mentioned above, the reduction in the STRF outlasted the excitatory region for up to ~10 ms relative to the onset of the excitatory signal component. Significance was assessed point-wise using t-tests, followed by the *Benjamini and Hochberg (1995)* algorithm for multiple comparisons applied to the p-values of the t-tests. At a false discovery rate of 0.01, the gray line shows the region of significant deviation.

Overall, the STRF analysis confirmed the presence of inhibition co-tuned with excitation, exhibiting a longer-lasting time-course of about 10 ms with respect to the onset of the excitatory signal component. Consequently, also under dynamic broadband stimulation, inhibition is confirmed to only marginally act above or below the neuron's excitatory receptive field and results in only a slight spectral sharpening of the SBC output. Next, we addressed the functional consequences of this co-tuned, prolonged inhibition.

## Glycinergic inhibition renders SBC responses sparser, more reliable and temporally more precise

The functional consequences of co-tuned inhibition appear less evident than those of narrow, sideband inhibition. The latter can diversely shape the response properties, by reducing responses only

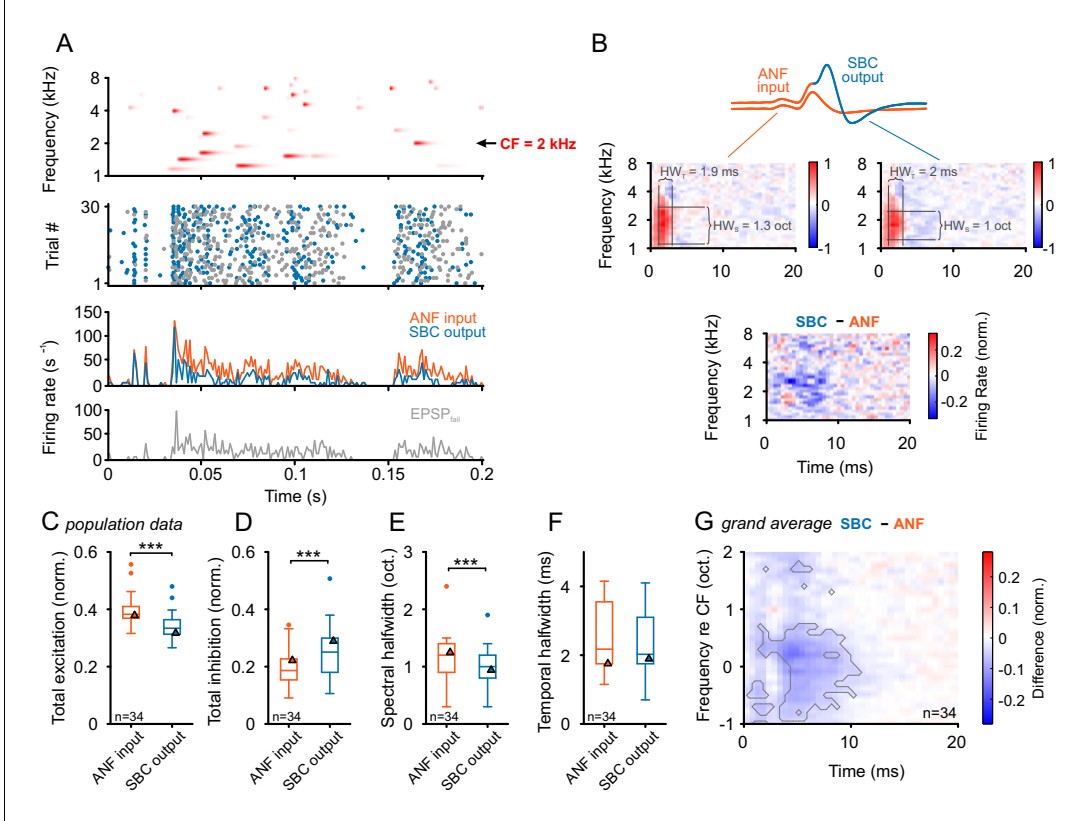

**Figure 7.** Input-output comparison of spectrotemporal receptive fields (STRF) indicates minor spectral sharpening and confirms broad, slow inhibitory action. (**A**) **Top panel:** Randomized gamma-tone sequence (RGS, scaling of red color indicates stimulus levels with a maximum of 70 dB SPL, see Materials and methods for stimulus details) were used to estimate STRFs of SBC output and its ANF input. The RGS spanned one octave below and two octaves above the unit's CF; in the present example 2 kHz. **Second panel:** Dot raster of discharges of an exemplary SBC evoked by 30 repetitive RGS presentations (blue = $EPSP_{succ}$, gray = $EPSP_{fail}$). **Third panel:** PSTH of the recording shown above; the graph differentiates between the total of the ANF input ($EPSP_{fail}$ + $EPSP_{succ}$, orange) and SBC output (SBC APs, blue). **Fourth panel:** From the same recording the histogram of the EPSPs that fail to trigger an SBC AP ($EPSP_{fail}$, gray). Note that EPSPs that elicited APs tended to be more prominent at the onset of excitatory response components. (**B**) STRF of the unit shown in A. **Upper panel:** Sketch of the two signal types, i.e. the totality of all EPSPs were considered to indicate the ANF input (orange), while EPSPs that generate an AP defined the SBC output (blue). **Middle panel:** corresponding STRFs. Note that there are clearly delineated areas of increased activity 2–3 ms after response-evoking stimulus components (red) which are distinct from areas with reduced activity. The spectrotemporal shape of the modulation at the ANF-SBC junction was quantified by the averaged difference-STRF. The STRFs of both ANF (**left**) and SBC (**right**) were computed separately and then subtracted (**bottom panel**). Relative temporal alignment was achieved by time-locking both ANF input activity and SBC output on the respective timing of maximum EPSP slope. The difference reveals changes in stimulus responsiveness in spectrotemporal coordinates. Negative values indicate a reduction in responsiveness, most likely caused by local inhibition. (**C–G**) Population data for all recorded SBCs (n = 34); triangles in the graphs indicate the respective values of the unit shown in A and B. (**C**) Stimulus-driven excitation was significantly reduced from the ANF input to SBC output, measured as the sum of all positive STRF bins (p<0.001, Wilcoxon signed rank test). (**D**) Stimulus-driven inhibition was significantly increased, measured by the negative sum of all negative STRF bins from the ANF input to SBC output (p<0.001, Wilcoxon signed rank test). (**E**) Spectral precision improved at the ANF-SBC junction, indicated by a reduced spectral half-width of the excitation (p<0.001 Wilcoxon signed rank test). (**F**) Temporal precision, estimated as the temporal half-width, was not changed between ANF input and SBC output (Wilcoxon signed rank test, p=0.16). (**G**) The average difference-STRF (n = 34 cells) exhibited a prominent and broad ( > 2 octaves) reduction around CF, which remained effective for ~10 ms (black line indicates significant deviation, adjusted for a false discovery rate < 0.01).

The following source data is available for figure 7:

**Source data 1.** Metrics of STRFs obtained with RGS stimulation.

for small, off-CF regions. Co-tuned inhibition, on the other hand, has been proposed to contribute to a precisely timed balancing of excitation to keep neurons within their dynamic ranges (*Renart et al., 2010*). To test for such a mechanism, we quantified properties of the SBC output in comparison to the ANF input with respect to the temporal sparsity of the response and

reproducibility across trials. Efficient neural codes have been proposed to show high sparsity, i.e. respond only rarely but then with high firing activity (*Field, 1994*). Again, the RGS stimulus was used to test the effect of acoustically evoked inhibition under complex acoustic conditions. The results yielded reduced mean firing rates of the SBC output compared to the ANF input (*Figure 8Ai*, data from an exemplary SBC) and increased sparsity in 28/32 cells (units above line of equality, *Figure 8Aii*). Sparsity was calculated by relating the variance of the neuronal response to its mean firing rate. The population analysis revealed significantly larger temporal sparsity in the SBC output than in its ANF input (ANF = 0.22 ± 0.08 vs. SBC = 0.31 ± 0.13, Δ = 0.09 ± 0.08, p<0.001, paired *t*-test, n = 34, U1 = 0.15, *Figure 8Aiii*, see also *Figure 8*–source data). Sparsity was calculated by relating the variance of the neuronal response to its mean firing rate (*Rolls and Tovee, 1995*; *Willmore and Tolhurst, 2001*), but other measures for sparsity yielded qualitatively similar results (see Materials and methods and Supporting *Figure 8*).

The reproducibility of the temporal response pattern was quantified by computing across-trial cross-correlations (*Figure 8B*). For this analysis, the obtained correlograms were divided by the product of the individual firing rates, rendering the results independent of absolute firing rates. Reproducibility was then calculated as the peak of the correlograms measured at 0 ms lag (*Figure 8Bi*). In 97% (33/34) of all recorded cells, the SBC output exhibited a higher level of reproducibility (units above line of equality, *Figure 8Bii*). Also, the population analysis yielded a significantly higher reproducibility of the SBC output than the ANF input (ANF = 0.46 [0.37, 0.65] vs. SBC = 0.83 [0.48, 1.3], Δ = 0.35 [0.12, 0.73], p<0.001, Wilcoxon signed rank test, n = 34, U1 = 0.18, *Figure 8Biii*).

To estimate the temporal precision across trials, the temporal dispersion was quantified as the half-width of the across-trial cross-correlation (*Figure 8Ci*). Temporal precision across trials improved from ANF input to SBC output in two-thirds of the recorded SBCs (24/34, units below line of equality, *Figure 8Cii*). Still, population analysis yielded a significant improvement in temporal precision (temporal dispersion: ANF = 7.04 [5.25, 7.93] ms vs SBC = 4.96 [3.47, 6.9] ms, Δ = 1.1 [0, 1.98] ms, p<0.01, Wilcoxon signed rank test, n = 34, U1 = 0.03, *Figure 8iii*). Response reproducibility across trials renders the response more identifiable for downstream processing stages which rely on precisely timed inputs. The increased sparsity reduces the energy expense by removing spikes which reflect the constant part of the response. Temporal precision of encoding also improved, although this was only observed in about 70% of the cells.

Finally, we directly tested whether the observed changes in response properties were indeed caused by acoustically evoked, glycinergic inhibition. Another set of 12 units were recorded under RGS stimulation, and glycinergic inhibition was blocked by iontophoretic application of strychnine (*Figure 8Aiv, Biv, Civ*). Like in the experiments reported above, the analysis differentiated between the ANF input to SBCs and the respective SBC output. Under control conditions, cells showed the above-described increase in sparsity and reproducibility at the ANF-to-SBC transition. Blocking the glycinergic inhibition resulted in decreased sparsity of the SBC output (*Figure 8Aiv*; control = 0.34 ± 0.1 vs. strychnine = 0.27 ± 0.07, Δ = 0.07 ± 0.06, p<0.01, two-way RM ANOVA, Bonferroni–adjusted, n = 12, U1 = 0.33). Also, the block of inhibition caused a decrease in response reproducibility (*Figure 8Biv*; control = 0.79 ± 0.39 vs. strychnine = 0.53 ± 0.19, Δ = 0.26 ± 0.23, p<0.05, two-way RM ANOVA, Bonferroni-adjusted, n = 12, U1 = 0.21) and an increase in temporal dispersion at the SBC output (*Figure 8Civ*; control = 5.6 ± 1.6 ms vs. strychnine = 7.5 ± 1.7 ms, Δ = 1.9 ± 1.6 ms, p<0.01, two-way RM ANOVA, Bonferroni-adjusted, n = 12, U1 = 0.21). In summary, the block of inhibition reduced the observed improvements from the ANF input to the SBC output, rendering both more similar. Importantly, the ANF input was not affected by the block of inhibition (*Figure 8Aiv, Biv, Civ*) (sparsity: control = 0.18 ± 0.05 vs. strychnine = 0.18 ± 0.05, Δ = 0 ± 0.01, p=0.5, U1 = 0.13; reproducibility: control = 0.32 ± 0.11 vs. strychnine = 0.32 ± 0.1, Δ = 0 ± 0.03, p=0.8, U1 = 0.08; temporal dispersion: control = 8.5 ± 0.9 ms vs strychnine = 8.5 ± 1 ms, Δ = 0 ± 0.6 ms, p=0.99, U1 = 0.08, n = 12, two-way RM ANOVA, Bonferroni-adjusted). In comparison with the pre-post data, the pharmacological dataset shows smaller variability across cells, which may be due to the lack of outliers in the latter, smaller dataset. In summary, these data directly show that glycinergic inhibition is a critical factor for the observed improvements from ANF input to the SBC output during complex acoustic stimulation.

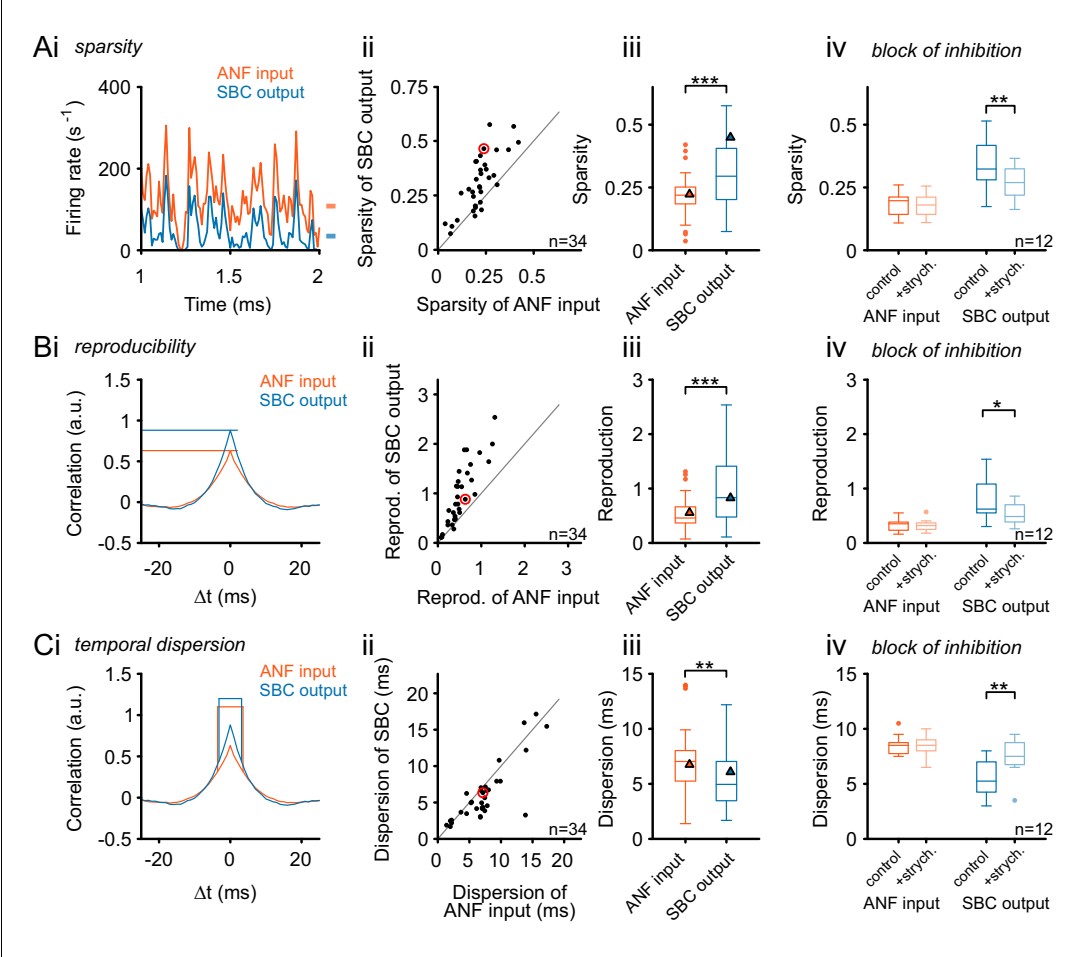

**Figure 8.** Inhibition renders the SBC responses sparse and increases across-trial reproducibility. (**A**) (**i**) Representative recording during RGS stimulation (2 s-section displayed) shows significantly sparser SBC output activity (blue) than the ANF input activity (orange). Marks on the right indicate the mean firing rate for ANF input and SBC output. (**ii**) Population data for all recorded SBCs (n = 34). Quantification of sparsity as the variance of the normalized firing rates shows that this relation holds for almost all units (dots above line of equality; red mark indicates representative unit on the left) and (**iii**) results in highly significant input-output differences (p<0.001, Wilcoxon signed rank test; triangle indicates the representative unit on the left). (**iv**) Blocking glycinergic inhibition in vivo by strychnine (n = 12) deteriorated the improved sparsity of the SBC output and rendered it similar to the ANF input (p<0.01, Wilcoxon signed rank test), while the ANF input remained unchanged. (**B**) The reproducibility of the response improved from ANF input to SBC output. Reproducibility was calculated as the time-aligned correlation between the neuron's responses to identical stimulus trials. High reproducibility indicates that the neural response is more constant across trials. (**i**) In the representative unit, higher reproducibility is seen for the SBC output (blue) compared to the ANF input (orange). (**ii**) Population data (n = 34) shows that the same relation holds for almost all units (data point marked in red indicates the unit shown on the left), and (**iii**) the statistical analysis yielded a high significant input-output difference (p<0.001, Wilcoxon signed rank; triangles indicate the respective values from the exemplary unit). (**iv**) Application of strychnine impoverished reproducibility in the SBC output (light blue) significantly compared to the control condition (dark blue; p<0.01, Wilcoxon signed rank test). The reproducibility of the ANF input (orange) was not influenced by blocking the inhibition (light orange). (**C**) The temporal dispersion for repetitive acoustic stimulation decreased from the ANF input to SBC output. The temporal dispersion was quantified as the half-width of the cross-correlation within each signal across trials (**i**). Population analysis showed improved temporal precision, i.e. reduced half-width/dispersion in the SBC output compared to the ANF input in most of the tested cells (**ii, iii**, same color coding as above, p<0.01, Wilcoxon signed rank test). As above, blocking inhibition increases temporal dispersion of the SBC output to the level of the ANF input (**iv**, p<0.01, Wilcoxon signed rank test).

The following source data and figure supplement are available for figure 8:

**Source data 1.** Sparsity, reproducibility and temporal dispersion for ANF input and SBC output.

**Figure supplement 1.** Alternative measures of temporal sparsity lead to consistent results.

## Subtractive inhibition suffices to explain the improvement in sparsity, reproducibility, and temporal precision

SBCs have been shown to be influenced by both hyperpolarizing and shunting effects of inhibition (*Kuenzel et al., 2011, 2015*; *Nerlich et al., 2014a*). While hyperpolarization has been attributed to a subtractive effect on firing rates (*Doiron et al., 2001*; *Silver, 2010*), shunting inhibition has mainly divisive effects (*Mitchell and Silver, 2003*; *Prescott and De Koninck, 2003*; *Capaday and van Vreeswijk, 2006*; *Ly and Doiron, 2009*). We investigated the functional effect of either type on the response via a simple simulation: either a fixed fraction (divisive, relative to the instantaneous firing rate) or a fixed number (subtractive) of spikes was removed from the ANF spike trains, matching the experimentally observed SBC output rates. Purely divisive inhibition, corresponding to a scaling of the PSTH, does not improve sparsity, reproducibility or temporal precision (*Figure 9*, purple, $\Delta$sparsity = 0 ± 0, p=0.99, $\Delta$reproducibility = 0.01 ± 0.02, p=0.25, $\Delta$temporal dispersion = 0.05 ± 0.41, p=0.89, n = 34, one-way RM ANOVA, Bonferroni-adjusted, see also *Figure 9—source data 1*). On the other hand, a purely subtractive inhibition matches the qualitative effects in the data well, i.e. improves all three properties (*Figure 9*, green, $\Delta$sparsity = 0.24 ± 0.08,

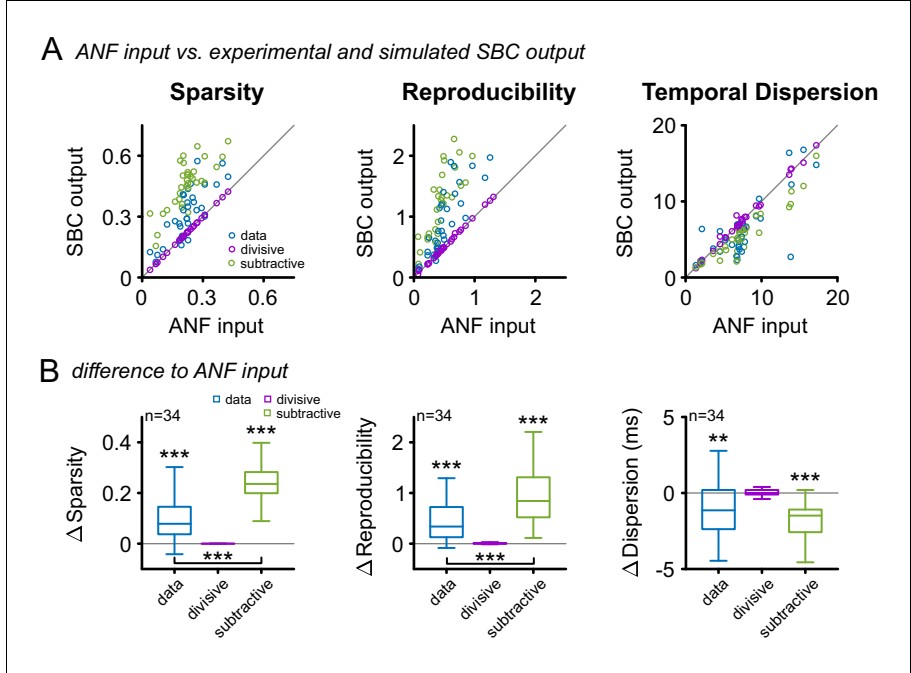

**Figure 9.** Subtractive inhibition, but not divisive inhibition can account for the improvement in sparsity, reproducibility, and temporal precision. (**A**) In response to the RGS stimulus, the SBC output (blue) showed a consistent increase in sparsity (left), reproducibility (middle) and decreased temporal dispersion (right). The simulated subtractive inhibition (green) showed similar improvements as the experimental data, while divisive inhibition (purple) had no effect on sparsity, reproducibility, and temporal dispersion. (**B**) These relations are also reflected in the population data, with significant changes in both the experimental data and the simulated subtractive inhibition (p<0.001, one-way RM ANOVA).

The following source data and figure supplements are available for figure 9:

**Source data 1.** Simulation of divisive and subtractive inhibition.

**Figure supplement 1.** Subtractive inhibition improves temporal precision and trial-to-trial reproducibility during SAM stimulation.

**Figure supplement 2.** Subtractive inhibition, but not divisive inhibition can account for the improved temporal precision and reproducibility during SFM stimulation.

p<0.001, Δreproducibility = 1.06 ± 0.73, p<0.001, Δtemporal dispersion = 1.8 ± 1.3 ms, p<0.001, n = 34, one-way RM ANOVA, Bonferroni-adjusted). Quantitatively, the simulated subtractive inhibition leads to larger improvements in sparsity and reproducibility than observed in the experimental data (*Figure 9*, blue, sparsity: data = 0.31 ± 0.13 vs. subtractive inhibition = 0.46 ± 0.12, Δ = 0.15 ± 0.07, p<0.001; reproducibility: data = 0.97 ± 0.61 vs. subtractive inhibition = 1.59 ± 0.9, Δ = 0.62 ± 0.6, p<0.001, n = 34, one-way RM ANOVA, Bonferroni-adjusted). We verified that similar relations hold for the SAM and SFM stimulation and the measures used in their analyses (*Figure 9*, *Figure 9—figure supplements 1* and *2*, respectively). A temporally unspecific, subtractive effect of inhibition might, therefore, be sufficient to explain the improvement in sparsity, reproducibility, and temporal precision. When combined with the divisive, co-tuned gain control, this improvement generalizes to a wide range of stimulus levels.

## Discussion

In the present study, we demonstrate that glycinergic inhibition shapes SBC responses to become sparser and more reproducible for a broad range of stimulation conditions. As a consequence, many temporal measures improve such as vector strength and across-trial temporal precision. We find inhibition to act largely co-tuned with excitation, although its latency and duration exceed the excitatory input, similar to the respective relationship found in the cortex. Therefore, we propose glycinergic inhibition to take a functional role as a gain control and a signal quality enhancer, which optimizes the SBC output for the subsequent high-fidelity integration for sound localization in the MSO and LSO (see below).

### Signal analysis and iontophoretic modulation confirm local inhibitory influence

The endbulb synapse depresses considerably during high-frequency firing (*Bellingham and Walmsley, 1999*; *Wang and Manis, 2008*; *Yang and Xu-Friedman, 2008*) despite the large size of the presynaptic synaptic terminal and the reliable, suprathreshold excitation observed in slice recordings. The present in vivo recordings showed that the increased failure fraction during acoustic stimulation cannot be explained by synaptic depression alone. This was evidenced by an analysis of EPSP thresholds and furthermore confirmed by iontophoretic application of a glycine receptor agonist and antagonist. In conclusion, the elevation of the EPSP threshold has proven to be a reliable indicator for inhibitory action, leading to an increased failure fraction. These data are consistent with previous in vivo studies (*Kuenzel et al., 2011*), as well as with slice and model studies demonstrating that an increase in inhibitory conductance can elevate threshold EPSP in bushy cells (*Xie and Manis, 2013*; *Kuenzel et al., 2015*). In summary, the endbulb of Held–SBC synapse seems to operate close to AP threshold and shows variable reliability which is strongly influenced by acoustically evoked inhibition (see also *Kopp-Scheinpflug et al., 2002*; *Kuenzel et al., 2011*; *Keine and Rübsamen, 2015*). While the observed frequency response areas are consistent with previous reports, we did not observe two distinct types of inhibition as reported earlier, i.e. broadband vs. on-CF inhibition (*Caspary et al., 1994*; *Kopp-Scheinpflug et al., 2002*). Instead, the present data suggest that inhibition at SBCs is broadband and on-CF.

### Iontophoretic application of glycine covers the physiologically relevant conditions

SBCs receive inhibitory inputs both on their somata and dendrites (*Gómez-Nieto and Rubio, 2009*). Both glycine and GABA receptors were shown to be present, with the latter playing a secondary role as demonstrated in slice experiments (*Nerlich et al., 2014a*, *2014b*). Therefore, we focused on the modulation of glycinergic inhibition. The applied dose was equated to match the acoustically evoked level of AP failures, keeping inhibition in the physiologically relevant range. This cautious approach will tend to underestimate the in vivo effect of glycine since the block of glycine receptors by local application of strychnine might be incomplete. Consistent with the slice data, the lack of threshold EPSP elevation during the block of glycinergic inhibition suggests only a minor influence of the GABAergic component during tone burst stimulation. The GABAergic inhibition might have an additional modulatory function or may only be activated during periods of high activity, as has been suggested for the glycinergic inhibition in the bird's nucleus magnocellaris (*Fischl et al.,*

*2014*), the avian homolog of the AVCN. Overall, we find glycine to have a substantial influence in shaping transmission at the SBC junction. However, the increased SBC output rates during block of glycinergic inhibition might increase the influence of spike depression (*Lorteije et al., 2009*; *Kuenzel et al., 2011*) and other factors such as spike threshold adaptation have to be taken into account (*Fontaine et al., 2014*; *Huang et al., 2016*).

## Inhibitory mechanism for improving sparsity and reproducibility of the neural response

For a broad range of acoustic stimuli, we observed a consistently sparser and more reproducible response in the SBC output compared to the ANF input. Can a simple inhibition achieve these changes in signal representation? Glycinergic inhibition has previously been demonstrated to be neither purely subtractive nor purely divisive (*Kuenzel et al., 2011*; *2015*; *Nerlich et al., 2014b*) and act with a short delay of ~3 ms on time scales of ~10–15 ms. These properties may be sufficient for increasing sparsity and reproducibility under the assumption that large deviations of firing rate are the consequence of stimulus-elicited, high firing probabilities rather than noise (see *Figure 9*). The latter would be temporally unrelated to the stimulus, and its transmission would reduce the reproducibility, and probably also the usefulness of the transmitted information for further processing.

In SBCs, this would translate to multiple, closely timed spikes for an individual input or across multiple excitatory inputs. The partially subtractive glycinergic inhibition would be strongly triggered by large instantaneous firing rates, and weaken subsequent inputs which do not occur closely to other inputs. On the PSTH level, a subtractive reduction thus almost inevitably increases sparsity (see *Figure 9*). Reproducibility will also be increased if the average temporal precision of high peaks is greater than the bulk of spikes at lower firing rates. Divisive inhibition typically leaves sparsity and reproducibility unchanged (*Figure 9*), although the influence on sparsity will partially depend on the measure used.

Due to the inhibitory/excitatory co-modulation with level, the enhancement in sparsity and reproducibility can extend over a wide range of levels. These considerations do not rule out additional enhancements of temporal precision, via additional excitatory inputs, however, these would be influenced similarly by the inhibition. The considerations above are challenging to study since precise temporal control over multiple inputs would be required.

Generally, if the information in the response is largely maintained, an increase in temporal sparsity can be advantageous for mainly three reasons. First, the information per spike is increased (*Barlow, 1972*, *2012*; *Olshausen and Field, 2004*). This is relevant for downstream cells, who can then combine information more efficiently with other inputs, to achieve higher precision, e.g. for estimating interaural time delays. Second, since the SBC's response occurs on a reduced plateau of unmodulated firing than the ANF's, changes in firing rate will lead to larger relative changes in firing rate, which should improve their detection in the target neurons. Avoiding saturating or strongly adapting postsynaptic responses also supports this improvement in change detectability (*Mitchison and Durbin, 1989*; *Graham and Willshaw, 1997*). Third, a reduction in the number of spikes reduces the energetic load on the system (*Levy and Baxter, 1996*; *Baddeley et al., 1997*; *Attwell and Laughlin, 2001*; *Olshausen and Field, 2004*; *Graham and Field, 2007*). In the present case the failures of transmission may, therefore, more appropriately be contrasted with the *spikes selected for transmission*.

## Inhibition at the endbulb synapse in the context of sound processing

SBCs provide the indispensable temporally precise excitatory inputs to the interaural time difference based sound localization in the MSO. Both, physiological and modeling studies suggest the neurons in the MSO act as coincidence detectors by primarily relying on the precise spike timing of binaural inputs (*Goldberg and Brown, 1969*; *Yin and Chan, 1990*; *Franken et al., 2014*). While sparsity has predominantly been advocated as an advantageous coding principle due to its reduced energy demand (*Levy and Baxter, 1996*; *Graham and Field, 2007*), in the MSO there might be a different justification for sparsity: With the MSO neurons acting as coincidence detectors (*Couchman et al., 2010*; *Plauška et al., 2016*), sparse excitatory inputs would lead to an improved signal-to-noise ratio of the correlation, i.e. of its peak in relation to the common floor of correlation. Presently, this is shown by the reproducibility measure across trials, which – when just considering

signal processing – can be regarded as the correlation between the activity of independent neurons from both sides (*Joris, 2003*). Equally important, the effective gain control achieved through the acoustically evoked inhibition will allow the MSO to retain a similar level of binaural sensitivity across a wide range of sound levels and stimuli. Importantly, this sparsening and increase in reproducibility were observed for the wide range of stimuli presently tested.

The observed increase in reproducibility in SBC output compared to the ANF input is consistent with previously reported population data. Studies comparing ANF and AVCN neurons (most likely spherical or globular bushy cells) in the cat reported increased reproducibility in neurons of the AVCN (*Louage et al., 2005*; *van der Heijden et al., 2011*). Here, we directly studied the increase across trials within single ANF-SBC junctions and not across cells. We conjecture that reproducibility across cells always increases if both cells increase their reproducibility individually. However, this can only be assessed with the peak of the cross-correlation, if the cells are matched in their tuning, a condition which is typically assumed for bilateral inputs into the MSO. For non-matched cells, reproducibility should increase but would have to be assessed with a different measure, which measures the set of responses to a given stimulus (for example Mutual Information).

Presently, we show that this increase is directly achieved at individual ANF-SBC junctions through the postsynaptic interaction of acoustically evoked excitation and inhibition. Also, during both narrowband and broadband stimulation, the block of inhibition resulted in a considerable decrease in reproducibility, sparsity, and temporal precision compared to the control condition (*Figure 8*). Hence, our results show that acoustically evoked inhibition plays a major role in shaping the SBC response. Most models of pitch perception are based on or related to an autocorrelation of neuronal activity and rely on temporal precision and reproducibility (*Licklider, 1951*; *Meddis and Hewitt, 1991*; *Cariani and Delgutte, 1996*; *Yost, 1996*; *de Cheveigné, 1998*; *Denham, 2005*; *Joris, 2016*). The improvement of both characteristics at the ANF-SBC synapse might support the neuronal processing of pitch, but experimental evidence is lacking. Other transmitters such as acetylcholine (*Fujino and Oertel, 2001*; *Goyer et al., 2016*) or norepinephrine (*Kössl and Vater, 1989*; *Rothman and Manis, 2003*) have modulatory effects on the SBC activity, but their effect on precisely-timed signal processing is not yet fully understood.

The output of SBCs is also relevant for general processing of sounds, i.e. their representation for later analysis, e.g. auditory recognition, a process distinct from sound localization (*Clarke et al., 1998*; *Maeder et al., 2001*). The increase in reproducibility might be beneficial in this part of the pathway as well. On the other hand, sparsity may have adverse effects, if the overall level is unavailable or if low level sound information in the ANF response is not represented anymore in the SBC response.

## Inhibition under spectrotemporally broad and dynamic stimulation

Natural stimuli tend to depart from the classical laboratory stimuli, in being spectrotemporally broad and diverse. We approximated this condition using the RGS stimulus in combination with distinct STRF estimation of both the pre- and the postsynaptic activity. STRFs were first introduced to study midbrain neurons in the grass frog (*Aertsen and Johannesma, 1980*; *Aertsen et al., 1980*, *1981*), and have since been an essential tool for auditory neuroscience along many stations of auditory system, ranging from the auditory nerve (*Kim and Young, 1994*) and the MNTB (*Englitz et al., 2010*) to the inferior colliculus (*Escabi and Schreiner, 2002*), and the auditory cortex (*Kowalski et al., 1996*; *David et al., 2012*). Recent developments in STRF estimation (*Theunissen et al., 2001*) make them a versatile tool to study the combined spectrotemporal stimulus selectivity of neurons for a wide range of acoustic stimuli. The present extension to pre- and postsynaptic activity is unique and provides a direct estimate of the spectrotemporal response modification occurring at a single endbulb synapse. It reveals inhibition to be co-tuned with excitation, but outlasting the latter, which – in total – leads to a slightly improved excitatory response precision. This response profile directly confirms the spectral properties gained from the pure-tone tuning curves and is in agreement with findings from of our previous studies on inhibition at the SBC (*Keine and Rübsamen, 2015*). The pre-post STRF analysis could provide a generally applicable tool for the investigation of a wider range of modulations of signal processing at the giant synapses in both MNTB and AVCN, for example the functional differences on synaptic transmission under the stimulation with natural acoustic stimuli or removal/application of specific neurotransmitters, e.g. GABA (*Nerlich et al., 2014b*) or neuromodulators, e.g. acetylcholine (*Goyer et al., 2016*).

The RGS stimulus allows a robust estimation of STRFs while keeping the spectrum sparse (unlike dense stimuli as the TORC stimulus, *Klein et al., 2000*). A sparse spectrum is advantageous for the present study of the potentially long-lasting inhibition (*Nerlich et al., 2014b*) since it prevents a continuous activation of inhibitory inputs causing saturation. It remains to be addressed, whether the RGS is a sufficient model for natural stimuli, or whether the natural statistics lead to a more specific activation of the inhibition arriving at the SBCs.

Overall level-dependence of STRF estimation was beyond the scope of this study and not investigated here in more detail. While the RGS stimulus contains different sound levels at different times and frequencies (via the randomized placement and shape of each gamma-tone), the overall, average sound level was kept constant. We predict that SBC STRFs will exhibit a greater robustness to changes in level compared to ANF STRFs, based on the gain-modulating inhibitory input (*Figure 3*).

## Source of inhibition

Several nuclei have been suggested to provide the inhibitory inputs to SBCs. The present data suggest the inhibitory source to feature a broad, symmetrically shaped tuning, consistent with an integration over a wide set of primary tuned inhibitory cells. The integration would have to be weighted by the distance to the postsynaptic CF, in order to achieve the symmetry in inhibitory modulation. In a recent study, *Campagnola and Manis (2014)* showed directly that bushy cells receive symmetric inhibition from within the CN. Further, the ~1 ms delay of the onset of acoustically evoked inhibition compared to the onset of excitation (*Kuenzel et al., 2011*; *Keine and Rübsamen, 2015*) suggests a single additional synaptic relay. Together, both the broadly tuned D-stellate cells in the AVCN and the tuberculoventral cells in the DCN are candidate sources (*Wickesberg and Oertel, 1990*; *Saint Marie et al., 1991*; *Campagnola and Manis, 2014*), as well as cells in the lateral nucleus of the trapezoid body (*Smith et al., 1991*; *Schofield and Cant, 1992*). The data of *Campagnola and Manis (2014)* directly demonstrate the CN as a source of inhibition, but neurons in other areas may contribute in addition. While we considered here the effect of the inhibition on stimulus representation at the SBC, this broad integration provides the opportunity for additional integration in the source areas.

In summary, while acoustically evoked inhibition on SBC renders the ANF-SBC junction less reliable with respect to signal transmission, it enables sparser and more reproducible SBC sound encoding, which might be of relevance for subsequent localization of sound sources.

## Materials and methods

### Animals and surgical procedure

All experiments were performed at the Neurobiology Laboratories of the Faculty of Bioscience, Pharmacy and Psychology of the University of Leipzig (Germany), approved by the Saxonian District Government, Leipzig (TVV 06/09), and conducted according to the European Communities Council Directive (86/609/EEC). Animals were housed in the animal facility of the Institute of Biology with a 12 hr light/dark cycle and with access to food and water ad libitum. Data were collected from 42 Mongolian gerbils (*Meriones unguiculatus*) of either sex aged 4 to 12 weeks (P43 ± 13).

Before the experiment, animals were anesthetized by an intraperitoneal injection of a mixture of ketamine hydrochloride (140 µg/g body weight, Ketamin-Ratiopharm, Ratiopharm, Ulm, Germany) and xylazine hydrochloride (3 µg/g body weight, Rompun, Beyer, Leverkusen, Germany). The surgical procedure was performed as described previously (*Keine and Rübsamen, 2015*). For multi- and single-unit recordings, the animal was tilted laterally by 12–18°. The recording electrode was lowered vertically by a step motor system into the anteroventral cochlear nucleus (AVCN). Glass micropipettes (GB150F-10 and GB150F-8P, Science Products, Hofheim, Germany) were fabricated with a PC-10 vertical puller (Narishige, Japan) to have impedances of 3–5 MΩ when filled with the pipette solution (in mM) 135 NaCl, 5.4 KCl, 1 $MgCl_2$, 1.8 $CaCl_2$, 5 HEPES, pH adjusted to 7.3 with NaOH. At the beginning of each recording session, multiunit recordings were performed with low-impedance electrodes (1–3 MΩ) to corroborate the stereotaxic coordinates of the rostral, low-frequency pole of the AVCN.

The activity of SBCs was then acquired by loose-patch recordings (*Lorteije et al., 2009*; *Kuenzel et al., 2011*). For that, the recording electrode was lowered through the cerebellum aiming

at the AVCN at a depth of about 5000 µm. When passing through non-auditory brain regions high positive pressure (200 mbar) was applied to prevent the electrode from clogging, and the electrode was advanced at a speed of 50 µm/s. On entering the target region, indicated by multiunit activity triggered by broadband noise search-stimuli, the pressure was reduced to 30 mbar, and the electrode then advanced in 1 µm-steps. When approaching a neuron, indicated by a gradual increase in series resistance, the pressure was equalized or slightly negative pressure (–5 mbar) applied. To minimize the mechanical stress on the recorded neuron, the seal resistance was kept <40 MΩ (*Alcami et al., 2012*). Single-units were recorded only when exhibiting a positive signal amplitude of more than 2 mV (dataset: 4.2 ± 1 mV) and a signal-to-noise ratio of at least 40 (mean amplitude of the positive AP peak divided by the standard deviation of the baseline, dataset: 68.2 ± 14.9).

## Iontophoretic application

To study the impact of inhibition on signal processing at the ANF-SBC synapse, loose-patch recordings were combined with iontophoretic drug application of the glycine receptor agonist glycine (Sigma-Aldrich, 100 mM, prepared in 0.9% NaCl, pH 6, buffered with 10 mM HEPES) and the glycine receptor antagonist strychnine (strychnine hydrochloride, Sigma-Aldrich, 5 mM, same formula). Three-barreled piggy-back electrodes (*Havey and Caspary, 1980*) were glued to the recording electrode and had the following steric configuration: tip diameter 4–8 µm, recording electrode protruding 20–40 µm (3GB120F-10, Science Products). The iontophoretic current was applied using an iontophoresis amplifier (EPMS-H-7 equipped with MVCS and MVCC modules, npi electronics) with increasing current steps ( + 5 to +100 nA). To reduce potential unspecific effects of strychnine, the application current was adjusted for each cell: First, the minimum current necessary to block spontaneous activity with glycine application was determined. Then, the iontophoretic current for strychnine application was set to block the glycine effect. Control experiments were performed by iontophoretic application of the carrier alone (0.9% NaCl, pH 6, buffered with 10 mM HEPES). Holding currents for each barrel was set to –20 nA and a channel filled with 0.9% NaCl was used for automatic capacitance compensation.

## Acoustic stimulation

All recordings were performed in a sound-attenuating and electrically isolated chamber (Type 400, Industrial Acoustics, Niederkrüchten, Germany) on a vibration-cushioned table. Acoustic stimuli were generated by custom-written Matlab software (MathWorks, Natick) and digitized at a rate of 97.7 kHz. Signals were presented via a custom-made earphone (DT48, beyerdynamic, Heilbronn, Germany) and delivered through a metal funnel ending just in front of the ear canal. The loudspeakers were calibrated using a condenser microphone (Bruel and Kjaer 4133) and custom-written Matlab software. Total harmonic distortions (ratio of the root-mean-squared (RMS) amplitude of higher harmonic frequencies to the RMS amplitude of the fundamental) were below 0.02% across all frequencies tested (0.1–40 kHz). All acoustic stimuli were corrected for the loudspeaker's impulse response prior to presentation.

## Data acquisition

Frequency response areas (FRA) were obtained by pseudorandom presentation of pure tones (100 ms in duration, 5 ms $cos^2$ ramps, 300 ms interstimulus interval) derived from a predefined matrix consisting of 20 different frequencies equally spaced on a log scale and 10 different sound pressure levels (SPL) equally spaced on a linear scale. Each of these 200 frequency/intensity pairs was presented 5–10 times while continuously recording the unit's discharge activity. The FRAs were used to detail each unit's CF, (the frequency at which the neuron is most sensitive), response thresholds, and – if present – the frequency-intensity domain of an inhibitory sideband.

### Sinusoidal amplitude-modulated (SAM) tones

Pure tones at the units' CF were amplitude-modulated at frequencies 50 Hz, 100 Hz, 200 Hz, and 400 Hz (200 ms duration, 500 ms interstimulus interval, modulation depth: 100%, starting at a phase angle of –90°).

### Sinusoidal frequency-modulated (SFM) tones

Tones were frequency-modulated in the range of one octave below to two octaves above the unit's CF at modulation frequencies 20 Hz, 50 Hz, 100 Hz, 200 Hz, and 400 Hz (duration: 200 ms, interstimulus interval: 500 ms).

### Randomized gamma-tone sequence (RGS)

Spectrotemporal receptive fields, sparsity, and reproducibility were estimated in response to a spectrotemporally broad and varied stimulus, a variant of the dynamic random chord stimuli (DRC) as described in *Ahrens et al. (2008)*. Briefly, the present DRC was generated by a randomized placement of equal-level gamma-tones in the spectrotemporal domain. Frequency locations were drawn independently according to a uniform distribution, relative to the CF of the cell, encompassing two octaves above and one octave below. The temporal separation between two adjacent Gamma-tones followed an exponential distribution with a time constant of 5 or 10 ms (see *Figure 7A* for an example). The bandwidth of the gamma-tones was varied along the frequency axis according to the model of *Zhang et al., 2001*, consistent with gamma-tone measurements along the Gerbil basilar membrane. The RGS was separately computed for each recorded unit and 20–30 identical repetitions of the 30 s long stimulus were presented while simultaneously recording the ANF input and SBC output.

## Data analysis

The rostral pole of the AVCN was targeted considering its tonotopic organization described previously (*Kopp-Scheinpflug et al., 2002*; *Dehmel et al., 2010*). Spherical bushy cells were recognized by their characteristic complex waveform (*Pfeiffer, 1966*; *Winter and Palmer, 1990*; *Englitz et al., 2009*; *Typlt et al., 2010*) and their primary-like PSTH pattern (*Blackburn and Sachs, 1989*).

The neurons' voltage signals were pre-amplified and impedance-converted (Neuroprobe 1600), A-M Systems, Sequim, USA), noise-eliminated (HumBug, Quest Scientific, North Vancouver, Canada), further amplified (PC1, Tucker-Davis Technologies), and digitized at a sampling rate of 97.7 kHz (24 bit, RP2.1, Tucker-Davis Technologies). Signals were band-pass filtered between 5 Hz and 7.5 kHz using a zero-phase forward and reverse digital IIR filter and stored for offline analysis using custom-written Matlab software.

Extracellularly recorded voltage signals of SBCs are typically composed of two (PP-EPSP) or three components (PP-EPSP-AP, *Figure 1*) reflecting the respective discharge of the presynaptic endbulb of Held (PP, prepotential), the postsynaptic EPSP, and the postsynaptically triggered AP (*Englitz et al., 2009*; *Typlt et al., 2010*). Signals were detected using a slope threshold for the rising flank of the EPSP. In this report, EPSPs that failed to trigger a postsynaptic AP were termed $EPSP_{fail}$ while EPSPs that successfully trigger an AP were termed $EPSP_{succ}$. The separation between $EPSP_{succ}$ and $EPSP_{fail}$ was based on the maximum falling slope following the detection time point. The APs following $EPSP_{succ}$ exhibited a considerably faster-falling flank than $EPSP_{fail}$ enabling a clear separation of both signal types (*Figure 1*). The detection thresholds were kept fixed for each recorded unit, but varied between units to account for different signal-to-noise ratios. For comparison of timing between $EPSP_{succ}$ and $EPSP_{fail}$, all events were time-stamped on their respective maximum EPSP slope. Previous studies showed that the maximum slope of the EPSP is a reliable measure of EPSP strength (*Kuenzel et al., 2011*). To determine the threshold EPSP, the maximum EPSP slopes of $EPSP_{succ}$ and $EPSP_{fail}$ were binned (bin size = 0.5 V/s), and the fraction of $EPSP_{succ}$ calculated for each bin. Then, a Boltzmann function of the form $\phi(x) = 1/\left(1 + e^{\frac{d-x}{a}}\right)$ was fitted to the data with each bin weighted relative to the number of events in that bin. The symmetric inflection point $d$ indicates the threshold EPSP, i.e. the EPSP slope necessary to yield a > 50% probability of triggering a postsynaptic AP (see also supplementary Matlab code). Earlier studies showed the influence of preceding neural activity on EPSC/EPSP and AP amplitude (*Englitz et al., 2009*; *Lorteije et al., 2009*; *Yang and Xu-Friedman, 2015*). While usually the preceding inter-event-interval (IEI) is used as a measure of previous activity, recent studies showed that short-term plasticity can extend well beyond the last IEI in vitro (*Yang and Xu-Friedman, 2015*). Therefore, a weighted average of all preceding events was used, with the impact dependent on the distance and EPSP slope of the respective events. The weighting was implemented as a single-exponentially decaying kernel, emphasizing temporally close events over more distant ones (*Sonntag et al., 2011*). For an EPSP slope at $t_0$ the

preceding activity was computed as $PreAct(t_0) = \frac{1}{median(S_i)} \sum_{i=-1}^{-\infty} S_i e^{(t_i-t_0)/\tau}$, with $S_i$ indicating the EPSP slopes and $t_i$ the temporal distance of the i-th preceding event. The time constant $\tau$ was set to 60 ms, as this was shown to be the time window of influence in slice studies (*Yang and Xu-Friedman, 2015*). In addition, the calculation was also performed with time constants of 10 ms and 100 ms yielding qualitatively the same results. The influence of preceding spiking activity on AP amplitude was performed in a similar manner, but limiting the preceding events to successful APs.

To evaluate the shapes of the inhibitory and the excitatory FRAs, asymmetry indices (AI) were calculated for both. AI was defined as $ln \frac{log_2(FU/CF)}{log_2(CF/FL)}$, where $CF$ indicates the neuron's characteristic frequency and $FL$ and $FU$ the respective low and high border-frequency of the FRA 40 dB above threshold (see also supplementary Matlab code). An AI of 0 indicates symmetric tuning curves, whereas negative and positive values describe asymmetric FRAs extending to lower or higher frequencies, respectively.

For sinusoidally modulated tones (SAM and SFM) the first 20 ms of every repetition were discarded from analysis to reduce the influence of onset effects, and analysis was constrained to complete periods to avoid unequal sampling. The temporal precision of spikes throughout the stimulus period was assessed by calculating the vector strength (*Goldberg and Brown, 1969*). The significance of phase-locking was tested based on the Rayleigh approximation (*Mardia, 1972*). Vector strength is an inadequate measure if the units firing rate reproduces the stimulus modulation. Therefore, we calculated the stimulus reproduction (Corr$_{Norm}$) as the normalized cross-correlation between the stimulus modulation and the respective response PSTH, adjusted for the latency of the neural response (see *Tolnai et al., 2008*) for a detailed explanation). Note that Corr$_{Norm}$ is constrained to the positive range [0, 1] and an unmodulated response would yield a value of 0.82. High Corr$_{Norm}$ values (>0.9) are only obtained when the response shape follows the stimulus envelope. The reproducibility of the neuronal responses was estimated by measuring the central peak of the shuffled autocorrelation across identical stimulus presentations (*Joris et al., 2006*). The modulation depth of the neural response to the 100% amplitude modulation was estimated by calculating the standard deviation of the first cycle of the normalized cross-correlation function.

## Estimation of spectrotemporal receptive fields

Spectrotemporal receptive fields (STRFs) represent the neural tuning in the dimensions of the spectrogram (time and frequency) and help to identify stimulus properties that control spiking at high temporal resolution. Specifically, we used STRFs to study the time course of inhibition during ongoing, spectrally dispersed stimulation. Estimation of STRFs was performed using generalized reverse correlation, as described elsewhere (*Theunissen et al., 2000*; *Englitz et al., 2010*). STRFs were estimated for both the input to the SBCs (EPSP$_{fail}$ + EPSP$_{succ}$), as well as the SBC output (EPSP$_{succ}$). Both input and output were aligned to their maximum EPSP slope, to enable a subtraction of the two STRFs using congruent reference points. STRFs were individually normalized to their standard deviation (positive peak lead normalization to very similar results), in order to allow an evaluation of tuning shape, removing the gain from overall firing rate. Without normalization, the result remains qualitatively the same. However, the inhibitory effect is then dominated by the difference at the STRF peak, which stems from the overall firing rate difference between ANF and SBC. The resulting difference-STRF indicates the translation in spectrotemporal sensitivity from ANF input to the SBC output (see *Figure 7*). We interpret this difference to be a consequence of three factors (i) local inhibition, together with (ii) postsynaptic processes of spike-frequency adaptation and (iii) Na-channel inactivation.

## Temporal sparsity, temporal precision and reproducibility of the neural response

We evaluated multiple measures of synaptic responsiveness to quantify the input-to-output signal processing at these second order neurons of the ascending central auditory pathway. All measures were computed separately for the ANF input and SBC output and then compared.

## Sparsity

The temporal sparsity of the neural response was calculated with three different methods, two classical and a third simple and intuitive one. First, we calculated the variance-based method introduced by *Rolls and Tovee (1995)* and *Willmore and Tolhurst (2001)*, with sparsity defined as

$$S = 1 - \langle r(t)_t^2 \rangle \,/\, \langle r(t)^2 \rangle_t$$

where $\langle . \rangle_t$ indicates an average over time. $S$ is an index ranging between 0 and 1. Second, we calculated the kurtosis of the firing rate distribution introduced by *Field (1994)*, which quantifies the peakedness of the firing rate distribution. Note, that this classical definition may lead to counterintuitive interpretations, e.g. if a neuron has dominantly high firing rates and is only rarely silent (also leading to high kurtosis). This measure is hence provided for historical reference.

Third, we calculated a simple and intuitive measure, which we term 'Close-to-Silence-Index' (CSI). The CSI is defined as the fraction of PSTH bins less than a certain firing rate $F$, i.e. $S = \{r(t)\}_t / T$, where $F$ is a firing rate threshold, chosen close to 0, and $T$ the total time of the PSTH. Different thresholds lead to different CSI values but allows one to define what a 'non-response' or 'small-rate' is. We here chose F = 15 Hz, although other sensible values (5–20 Hz) gave qualitatively similar results. The CSI is a useful estimator if the firing rate distribution is monomodal. Sparsity analysis was performed on PSTHs sampled at 100 Hz (see also supplementary Matlab code).

## Reproducibility

Across multiple repetitions of a stimulus, the neural response can repeat reliably or exhibit a high trial-to-trial variability. In the present experiment, variability on the stimulation side is marginal. Thus, the observed variability is solely due to the neural processing. We calculated a measure of reliability by computing the cross-correlation across different trials (same trials always excluded) in response to the same stimulus, similar to the correlation index (*Joris et al., 2006*). The height of the central peak of correlation was termed reproducibility (see *Figure 8C1* for an illustration and supplementary Matlab code).

## Temporal precision

Lastly, the temporal precision of SBC AP generation was quantified by the half-maximum width of the central peak of the cross-correlation function across identical stimulus presentations. The width is termed dispersion, measured in milliseconds (see *Figure 8B1* for an illustration). The slimmer the central peak, the higher the temporal precision of neural activity in representing complex acoustic stimuli.

## Statistics

Data sets were tested for Gaussianity and equality of variance using the Shapiro-Wilk test (*Shapiro and Wilk, 1965*) and Levene's test (*Levene, 1960*), respectively. Comparison between two independent groups was performed using student's two-tailed *t*-test or Wilcoxon rank sum test as appropriate. Aggregated data are reported as mean ± standard deviation or median [first quartile, third quartile], respectively. Within-subject comparisons were performed by paired *t*-test, Wilcoxon signed rank test, or by multi-factorial repeated-measures (RM) ANOVA after testing for sphericity using the Mauchly test (*Mauchly, 1940*). If the assumption of sphericity was violated, Greenhouse-Geisser correction was applied. Bonferroni correction was applied to all multiple comparisons (*Bonferroni, 1936*). Correlation between quantities was assessed by Spearman's rank correlation (*Spearman, 1904*) to cover linear and nonlinear relationships. For interpretation of all results, a p-value less than 0.05 was deemed significant. Significance thresholds are abbreviated in figure panels as asterisks, with *, **, ***, corresponding to p<0.05, p<0.01, p<0.001, respectively. The effect size was calculated using the MES toolbox in Matlab (*Hentschke and Stüttgen, 2011*) and reported as eta-squared ($\eta^2$) for RM ANOVA and Cohen's U1 for two-sample comparisons. No statistical methods were used to pre-determine sample sizes.

## Acknowledgements

This work was supported by the DFG grants GRK 1097 (CK), RU 390/19–1, RU 390/20–1 (RR), and Marie Sklodowska Curie Fellowship 660328 (BE). We thank Jörg Encke for helpful discussion and Sebastian Maass and Ingo Kannetzky for technical support. We thank the three anonymous reviewers for their helpful suggestions which substantially improved the manuscript. The authors declare no competing financial interests.

## Additional information

### Funding

| Funder | Grant reference number | Author |
|---|---|---|
| Deutsche Forschungsge-meinschaft | GRK 1097 | Christian Keine |
| Deutsche Forschungsge-meinschaft | RU 390/19-1 | Rudolf Rübsamen |
| Deutsche Forschungsge-meinschaft | RU 390/20-1 | Rudolf Rübsamen |
| European Commission | Marie Sklodowska Curie Fellowship 660328 | Bernhard Englitz |

The funders had no role in study design, data collection and interpretation, or the decision to submit the work for publication.

### Author contributions

CK, Conception and design, Acquisition of data, Analysis and interpretation of data, Drafting or revising the article; RR, Conception and design, Drafting or revising the article; BE, Conception and design, Analysis and interpretation of data, Drafting or revising the article

### Author ORCIDs

Christian Keine, http://orcid.org/0000-0002-8953-2593
Bernhard Englitz, http://orcid.org/0000-0001-9106-0356

### Ethics

Animal experimentation: All experiments were approved by the Saxonian District Government, Leipzig (TVV 06/09), and conducted according to the European Communities Council Directive (86/609/EEC).

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
