## [Decision Letter]

Thank you for submitting your article "Inhibition in the auditory brainstem enhances signal representation and regulates gain in complex acoustic environments" for consideration by *eLife*. Your article has been reviewed by three peer reviewers, one of whom is a member of our Board of Reviewing Editors and the evaluation has been overseen by Andrew King as the Senior Editor. The reviewers have opted to remain anonymous.

The reviewers have discussed the reviews with one another and the Reviewing Editor has drafted this decision to help you prepare a revised submission.

All reviewers agreed that this paper was an interesting study that enhanced our understanding of the transformation of information between the auditory nerve and the cochlear nucleus. The data were generally well analysed (but see below for suggestions for improvement) and no further experiments were deemed necessary. While some of the results in the early part of the paper have been reported by others (including their own group), the latter part of the paper, looking at the coding of more natural stimuli i.e. SAM, SFM and RGS, contains new and interesting data.

There are quite a few comments, but the reviewers feel these can all be satisfactorily addressed by the authors. In particular, the reviewers would appreciate a clearer explanation of the STRF stimulus and its analysis e.g. concepts such as sparsity and reproduction need to be made more accessible. In addition, the authors should address the issues of data analysis and presentation outlined in the comments below.

Essential comments

1) It is important to set out more clearly what this study is addressing. To say simply that "the functional role [of inhibition] is not entirely understood" (Introduction section) is inadequate. In the Abstract and Introduction you suggest that one possibility is to compensate for stimulus level. The passage at the beginning of subsection “Inhibition at the endbulb in the context of sound localization” seems important to this argument, but it is difficult to understand how sparsity enhances coincidence detection in MSO. If two inputs simply fire less, then there are fewer opportunities for coincidence, so it seems coincidence detection could get worse if the spikes are deleted at random. The authors posit that reproducibility also occurs across cells, not just within. Do the present data support that? How does level tolerance arise out of this?

2) The concepts of sparsity and reproduction need to be more clearly explained. This is especially true in subsection “Glycinergic inhibition renders SBC responses sparser, more reliable and temporally more precise”, but even in the Abstract, these terms should be clearly defined.a) Why is kurtosis a measure of sparsity? Kurtosis seems to capture both tails of a distribution, but the term "sparsity" implies decreased firing generally, which would include a shift in the mean firing rate, or a skew towards longer intervals. Kurtosis, as the 4th moment, seems to be chosen to avoid both those features. Why is that? It should also be clearly explained why sparsity might be important. The argument from saving energy is not very compelling, if there is not also a decrease in firing rate.b) The term reproduction is not clear. Its definition is hidden in the Methods and while the values are compared in Figure 5, an example is not shown until Figure 8. "Reproduction" has a connotation of being able to recreate something (e.g. using a response to reproduce the stimulus). However, the quantification seems to be using cross-correlation of responses. Perhaps "reproducibility" would be a better term for this measure, though it is important to consider that just because a response is reproducible does not mean it is correct or efficient or useful.

3) Do all primary-like units show failures? Did you just select cells that showed failures? Please comment on cell type. How is it determined that these are spherical bushy cells? Are recordings targeted towards cells with particularly large synaptic inputs? How many synaptic inputs does each cell receive? Are the electrical signals assignable to activity of one input? Does the analysis of Figure 2 assume these are single inputs? If they are multiple synaptic inputs, how is this approach justified?

4) Spike depression. Lorteije et al., 2009 suggested that "spike depression" (presumably the refractory period) contributed to EPSP failures at the calyx of Held. The refractory period would likely appear as an increase in threshold during periods of high activity, similar to Figure 2 right, and a decrease in spike amplitude with high activity, similar to Figure 2 right. This possibility should at least be acknowledged, and the strychnine results interpreted with this in mind as well.

5) Second paragraph subsection “Synaptic depression alone fails to fully account for increased failure rates”: Mathematically, what you have done here is correct, but it will likely be confusing to many readers. It is unclear why an "error function" (Boltzmann?) is used to fit this data, as the points clearly cluster into two clouds, and the "threshold" point is not near any of the data points. Why were the data not binned along the ordinate of rising slope (in 0.5 or 1 V/s bins), and the probability of success or failure computed in each bin, based on the total number of succ+fail. This would give the type of representation that would then make sense in terms of fitting to a Boltzmann to obtain a "threshold". Maybe do this as an inset?

6) The summary plots can be very difficult to read. Each plot shows median, quartiles and outliers, as well as averages. This gets very busy, and winds up obscuring or minimizing the important differences. For example, in Figure 3Ciii the averages are squashed down in the lower part of the panel, and most of the panel is taken up with outliers. It would be better to maximize the plot area to make the data as visible as possible. Averages and standard errors are sufficient when testing with a t-test, as are box-and-whisker plots when a non-parametric test. This applies to summary plots throughout, with Figure 5 particularly hard to read.

7) Many plots compare paired values, and report the original values (+/- SE), but not their differences. For example, 3Ci reports average CFs of ANF and SBC. The average values of these are not very meaningful. What is important is the difference between CFs for input vs. output, which is not reported. This applies in many other places: in Figure 3, differences in threshold, Q, asymmetry; in Figure 4 right panels, effects of glycine, strychnine, stimulation; in Figure 5 and Figure 6, differences between input and output; Figure 7, differences between input and output, Figure 8iii, iv, differences between input and output, effects of strychnine.

8) ANOVAs in Figure 5, 6. It is very difficult to understand the ANOVA carried out here. In Figure 5, left, the ANOVA results suggest there is "a systematic variation with modulation frequencies". This is not visible in the plot, and furthermore, the nature of the "systematic variation" is obscured by an ANOVA. Did it go up or down? In the FM analysis in Figure 6, the extreme frequencies show little effect, but 100 Hz modulation shows a strong effect. Why is this behavior so non-linear? Did the results match some sort of expected outcome?

9) Flatness of the RLF (Figure 3). Using standard deviation to quantify flatness of the RLF is confusing. A standard deviation is a comparison to a mean, and the mean here is not meaningful because it depends on which intensities were sampled. My sense is that the standard deviation would depend on whether the RLF is sampled symmetrically about a cell's dynamic range (the SD would be maximal), or asymmetrically (the SD would be lower). Please justify this measure more carefully. A more intuitive measure of flatness may be the difference in firing rate between maximum and minimum (normalized appropriately).

10) Effects of inhibition. The analysis of the effects of inhibition is very thorough, and beautifully shown, but does not adequately acknowledge similar results from earlier work. This is particularly noticeable in the final paragraph of subsection “Acoustically evoked inhibition elevates threshold for AP generation”, which does not acknowledge that Kopp-Scheinpflug, 2002 and Kuenzel, 2011 also showed similar tuning of inhibition. Figure 5–Figure 7 investigate different types of stimuli, but it would be helpful to compare the findings against others' results using tuning curves. The period histograms in Figure 5 and Figure 6 seem to show that the output is simply shifted lower compared to input. Is such a simple overall shift sufficient to explain the changes in vector strength, modulation depth, reproduction, and correlation? Or, are the temporal characteristics of inhibition also important?

11) FM data period histograms. The FM stimuli appear to yield 2 or 3 peaks in the representative example of Figure 6. Multi-peaked firing can undermine the utility of the standard VS calculation. That is, each peak may be very tight and repeatable, but by having more than one, the VS decreases.

12) STRF analysis: This analysis is very nice, but some clarification would be helpful.a) Why is the inhibitory STRF calculated by subtracting the ANF and SBC plots? Please explain how and why firing rates are normalized by the standard deviation. Is it normalized by subtraction or division? What if this normalization is not done? Please add a scale bar to these plots. Can this analysis be done instead by using only the EPSP(fail) for the kernel estimation? How would this approach differ from the subtraction approach?b) The size of the excitatory region is described as sharper in the SBC compared to ANF. This is not obvious in the example. Can the spectral width be indicated on the plot in Figure 7 somehow? Is there a similar effect in the tuning curves in Figure 3? It is not clear how sound level influences the STRF. Would that affect this width measurement?

13) Figure 8: This is a very important figure, because it addresses how inhibition influences the information carried by SBCs compared to ANFs. It would be very helpful if the authors could take a stab at explaining how inhibition may enhance sparsity, reproduction, and temporal dispersion. In the case of sparsity, it seems like it cannot be simply a change in the mean firing rate, because kurtosis should not include that (unless there are weird effects of rectification at firing rate of 0).

---

## [Author Response]

*[…] Essential comments*

*1) It is important to set out more clearly what this study is addressing. To say simply that "the functional role [of inhibition] is not entirely understood" (Introduction section) is inadequate. In the Abstract and Introduction you suggest that one possibility is to compensate for stimulus level. The passage at the beginning of subsection “Inhibition at the endbulb in the context of sound localization” seems important to this argument, but it is difficult to understand how sparsity enhances coincidence detection in MSO. If two inputs simply fire less, then there are fewer opportunities for coincidence, so it seems coincidence detection could get worse if the spikes are deleted at random. The authors posit that reproducibility also occurs across cells, not just within. Do the present data support that? How does level tolerance arise out of this?*

We reckon that the aims and insights of the study could have been more clearly described early on. The review process has motivated us to investigate the mechanisms deeper than before (see below), which leads to a clearer and more insightful bottom-line:

" These improvements are a consequence of the combined subtractive/divisive action of glycine (Kuenzel et al., 2011, 2015): The subtractive component enhances the temporal sparsity by raising the threshold for spiking. Therefore, mostly high instantaneous firing-rate events pass the junction, which often correspond to salient events in the cell's frequency range. The divisive component acts primarily as a gain control, which – in conjunction with the co-tuning – maintains the SBC output rate in a smaller range across different stimulus levels. Together these two effects focus the SBC output onto well-timed stimulus events across a wider range of stimulus levels. Thus, inhibition improves the basis for the high-fidelity signal processing in downstream nuclei crucial for sound localization irrespective of the prevailing stimulus levels."

We have added this passage in the Introduction section, in addition to a number of other clarifications.

Regarding the other questions raised in this point:

The reviewers correctly point out that the step from improved sparsity and reproducibility to improved coincidence detection is not a trivial one, and should be addressed more directly. As the reviewers indicate, a random deletion of spikes would not lead to improved coding. Since a random deletion would correspond to a divisive scaling of the PSTH, it would also not lead to an increase in sparsity (although this would depend to some degree on the measure, see the next point). The effect of inhibition in this case is at least partially subtractive (see also #10 and #13 of the essential comments below for more details), which means that the spikes/parts of the PSTH that remain are those occurring during phases of high firing rate or high firing probabilities. These are the phases, when the stimulus drives a lot of spikes in the current frequency band/neural tuning (directly shown e.g. by the STRF tuning). For a given excitatory frequency these phases will correspond to high stimulus level (given the monotonic rate-level functions of ANFs), which are likely to be present in both the left and the right ear. We propose, that in this way, a given frequency channel transmits the most salient events in its frequency channel, which will also have the best timing (based on the general level-latency dependence (Heil and Irvine, 1997; Heil et al., 2008; Neubauer and Heil, 2008). Such an increase in temporal precision was also demonstrated before, (e.g. (Joris et al., 1994a, 1994b; Dehmel et al., 2010; Kuenzel et al., 2011) and also our Figure 5, Figure 6 and 8. Note, that due to the co-tuning of the inhibition with excitation, the strength of inhibition varies across level. Thus, across all levels the neuron’s response to the *relatively* loudest peaks are kept, which is what we refer to as gain control.

Next, the outputs from both ears are then combined in the MSO (and LSO) for sound localization: In order to precisely estimate sound source locations, MSO neurons have to detect “correct” coincidences, i.e. coincident spikes from both ears while neglecting coincident spikes from one ear or coincidences that are purely random. MSO neurons possess a number of anatomical (Stotler, 1953; Smith, 1995; Agmon-Snir et al., 1998) and biophysical specializations (Scott et al., 2007; Mathews et al., 2010) to distinguish between monaural and binaural coincidences. The increase of ANF firing rates with sound pressure level would – at high sound intensities – likely result in a large number of coinciding inputs at the MSO level and create MSO activity at unfavorable ITDs, i.e. coincident spikes just by chance alone. The above described focus on the relatively highest-level events may allow the MSO to focus on binaural spikes with high temporal fidelity, and thus avoid inaccurate contributions to the ITD estimation. The increase of inhibition with level maintains a more limited range of firing rates across level. Given the presumably strong biophysical basis of coincidence detection in the MSO, it seems plausible that precise ITD processing can only function over a limited range of input firing frequencies. However, as intuitive as this reasoning may seem, it remains partly speculative at this point. We have added considerations along the lines above in the Discussion, adding to Discussion sections that have been added in response to other points raised by the reviewers (see section: Inhibitory mechanism for improving sparsity and reproducibility of the neural response)

Next, the reviewers inquired whether reproducibility increases across cells as well. Before answering this point, we would like to make our claim more precise: If reproducibility is measured with cross-correlation at time-shift 0, an increase in reproducibility in a single cell, does not necessarily lead to an increase in reproducibility across multiple cells. Since different cells will have different tuning properties, their cross-correlation may well reduce (at time-shift 0), even if both cells individually increase their cross-correlation at time-shift 0. If cells with matched tuning properties are compared, however, an increase in reproducibility in each cell, should translate to an increase across cells. We have tested this directly for a subset of cells, identified by clearly peaked ANF cross-correlations (Figure 10). We selected cells where identical stimulus parameters were used and in all tested cases (14 pairs of cells) the cross-correlation at time-shift 0 increased, as predicted. However, although the increase across cells was positive (0.15+/-0.18, Wilcoxon signed ranks test, p<<0.001) it was slightly smaller (Wilcoxon signed ranks test, p=0.03) compared to the increase within the same set of cells (0.53+/-0.41), which could be a consequence of the residual tuning mismatch between these cells.

Author response image 1.Reproducibility of the SBC output also improves across cells, if these are well matched in their CF and stimulus.Top/right: Cross-correlograms between ANF input (orange) and SBC output (blue) across pairs of cells (columns and rows enumerate the cells, diagonal is the same cell (gray background), for comparison). Our sample contained 9 cells (one group of 3, and one of 6 cells), which were stimulated with the identical RGS stimulus, giving a total of 18 across pairs. One of the cells did not show an improvement in reproducibility (with itself) and correspondingly did also not show an improvement with other cells (blue background). Since the reviewers asked about improvements in reproducibility, this cell is excluded from the population analysis below. Bottom left: Reproducibility between ANF input and SBC output within (purple) and between (black) cells. Reproducibility also increased between different cells with similar tuning properties (dots above line of equality). Bottom middle: Population data indicate a significant increase in reproducibility of the SBC output compared to the ANF input between cells (p<0.001, n=14, Wilcoxon signed rank test.**DOI:**
http://dx.doi.org/10.7554/eLife.19295.024

On the other hand, if reproducibility were measured with a more general measure, e.g. the entropy of the response given a stimulus (i.e. H(R|S)), or – more typically – mutual information (I(R,S) = H(R)-H(R|S)), then we would predict that an increase in reproducibility on single cells would always lead to an increase in reproducibility on the population level. While the current number of repetitions would make entropy estimation inaccurate (Panzeri and Treves, 1996), the general argument for an overall increase is as follows: considering the extreme case, where every cell produces an individual, but unique spike-train for a given stimulus. Then the neural population would also emit a unique population spike-train, thus be highly reproducible. Similarly, if a single cell becomes more reproducible, this reduces the set of response emitted by the entire population. We have added the following abbreviated version of this argument to the Discussion:

“Here, we only directly studied the increase across trials within a cell, but not across cells. We conjecture that reproducibility across cells always increases, if both cells increase their reproducibility individually. However, this can only be assessed with the peak of the cross-correlation, if the cells are matched in their tuning, a condition which is typically assumed for bilateral inputs into the MSO. For non-matched cells, reproducibility should increase, but would have to be assessed with a different measure, that measures the set of responses to a given stimulus (for example Mutual Information).”

*2) The concepts of sparsity and reproduction need to be more clearly explained. This is especially true in subsection “Glycinergic inhibition renders SBC responses sparser, more reliable and temporally more precise”, but even in the Abstract, these terms should be clearly defined.a) Why is kurtosis a measure of sparsity? Kurtosis seems to capture both tails of a distribution, but the term "sparsity" implies decreased firing generally, which would include a shift in the mean firing rate, or a skew towards longer intervals. Kurtosis, as the 4th moment, seems to be chosen to avoid both those features. Why is that? It should also be clearly explained why sparsity might be important. The argument from saving energy is not very compelling, if there is not also a decrease in firing rate.*

We added additional information on sparsity and reproducibility to the Introduction and Results section at their first appearance. The word limit in the Abstract prevented us from explaining the concepts in more detail there.

We used the kurtosis of the firing rate distribution to measure the sparsity of the neural response, since it had been classically suggested as a measure of sparseness (Field, 1994) with higher values of kurtosis related to higher sparseness (see Figure 6 of Field, 1994). The rationale for the use of kurtosis in the original work was to assess the 'peakedness' of the firing rate distribution, which led Field to consider kurtosis as a "useful measure", although not necessarily the best measure for sparseness.

We, however, agree with the reviewer's observation, that the use of kurtosis is too general in the present context, since both tails above and below the average contribute, hence, there could be cases with identical kurtosis but very different average firing rate. We, therefore, apply another method of sparseness quantification, which does not suffer from this problem. This measure has been suggested by Rolls and Tovee, (1995) and later taken up by Willmore and Tolhurst (2001), as well as several subsequent studies in neuroscience (~200, according to citation counts on Google Scholar).

This measure is defined as: S=1−r(t)t2 / r(t)2t , where .t denote the time average. S is close to 1, i.e. maximal sparseness, if the square of the average rate ([r(t)]) is very small compared to the squared deviations of the rate from zero ([r(t)2]).

On the other hand, S is close 0, if the average rate is high, and the variations in rate comparatively low.

Note, that from the definition of the Var(r(t))=[r(t)2]−[r(t)]2>0, it can be deduced that the denominator in S is always greater or equal the enumerator. Hence, S varies only between 0 and 1, and therefore empirical differences in between S of ANF and SBC responses can be compared against 1.

As an alternative 'safe & clean' measure, we introduce another measure of sparseness, here, termed 'Close-to-Silence-Index' (CSI). CSI is defined as the fraction of PSTH bins, which are below a certain threshold (here chosen to be 15 Hz, quite low compared to the range of up to 400 Hz firing rate, see Figure 8—figure supplement 1), i.e. CSI=#Binst(r(t)<Threshold)/#Binst.

CSI is also an index, thus ranges only between 0 and 1. CSI essentially directly evaluates how close a cell is to spiking at low rate for most of the time. Assuming monomodal firing rate distributions, which are limited from below by 0, it should be a reliable measure for sparsity as well, especially if a conservative threshold is chosen.

Overall the results between the three sparseness measures do not differ qualitatively, all indicating the SBC responses to be temporally sparser than the ANF responses. Since kurtosis has historically been used we keep it in the supplement figure together with the CSI, and replace the – to our knowledge – most standard definition of sparseness by Rolls and Tovee, 1995 in Figure 8.

Secondly, as suggested by the reviewers, we now address the virtues of sparse coding in greater detail in the Introduction (briefly) and the Discussion (at greater length), but not in the Results, since this remains an interpretation. Concretely, we focus on the information per spike and the energy efficiency: One main advantage of sparsity can be the increased information per spike, e.g. to temporal information, or also the detection of changes in firing rate/stimulus properties, if sparsity is achieved through gain normalization. Since the firing rate does indeed decrease in all cells (after all we are studying failures of transmission), we also keep the argument of more energy-efficient coding, but provide a more detailed argument now.

*b) The term reproduction is not clear. Its definition is hidden in the Methods and while the values are compared in Figure 5, an example is not shown until Figure 8. "Reproduction" has a connotation of being able to recreate something (e.g. using a response to reproduce the stimulus). However, the quantification seems to be using cross-correlation of responses. Perhaps "reproducibility" would be a better term for this measure, though it is important to consider that just because a response is reproducible does not mean it is correct or efficient or useful.*

We agree that “reproducibility” is more in line with our definition (and maybe more grammatical) than “reproduction”, hence, we replaced it throughout the revised manuscript. To introduce the term an explanation has been added to the Results section and a subplot has been added to its first mention in Figure 5 which illustrates its definition.

Further, we also agree that increased reproducibility of the neural response does not necessarily imply a more efficient or useful coding of acoustic information (for 'correct', see below). However, reproducibility across trials, which we measure here, provides a basis for the consistency in extracting information (e.g. interaural time difference, but also stimulus envelope) to repeated presentations of the same stimulus. This is the minimal requirement one should have for a consistent processing of stimuli.

According to Joris, 2003, comparing each pair of spike trains from recordings of a single cell can be considered as providing the input to a simple binaural coincidence detector. While we think the quantification on the basis of the cross-correlation is generally sensible, it is particularly fitting in the context of subsequent computations for extracting interaural time-delays in subsequent processing stations, which likely relies on operations similar to cross-correlation (Colburn, 1996; van der Heijden et al., 2013; Franken et al., 2014; Plauška et al., 2016). For example, MSO neurons are thought to perform an approximate cross-correlation between their bilateral inputs, hence, an increase in cross-correlation translates into a more precise firing of MSO neurons, i.e. more precise in interaural time.

The last point raised by the reviewers, whether a more reproducible response is more correct, is hard to answer, without relating it to the subsequent computation. For example, if only interaural-time difference were computed, then the combination of increased reproducibility and temporal precision can be taken as a signature of a 'more correct' response for this purpose. However, this could also render the response less correct for other purposes. We have added these important considerations to the Discussion. They are embedded into the section dealing with functional origin of the increase in sparsity and reproduction (see below):

"These properties may be sufficient for increasing sparsity and reproducibility under the assumption that large deviations of firing rate are the consequence of stimulus-elicited, high firing probabilities rather than noise (see Figure 9). The latter would be temporally unrelated to the stimulus, and its transmission would reduce the reproducibility, and probably also the usefulness of the transmitted information for further processing […]."

*3) Do all primary-like units show failures? Did you just select cells that showed failures? Please comment on cell type. How is it determined that these are spherical bushy cells? Are recordings targeted towards cells with particularly large synaptic inputs? How many synaptic inputs does each cell receive? Are the electrical signals assignable to activity of one input? Does the analysis of Figure 2 assume these are single inputs? If they are multiple synaptic inputs, how is this approach justified?*

From our recordings at the rostral pole of the AVCN, all cells showed failures but to very different degrees (as shown in Figure 1). We did not encounter a cell that was completely fail safe and even for cells which showed low failure rates during spontaneous activity, the failure rate generally increased under acoustic stimulation. The SBCs were determined by a number of physiological properties such as high-spontaneous rates (Smith et al., 1993), localization at the rostral pole of the AVCN (Bazwinsky et al., 2008), the presence of a discernible prepotential in addition to complex waveform (Pfeiffer, 1966; Englitz et al., 2009), short AP duration (Typlt et al., 2012), and primary-like response patterns to pure-tone stimulation (Blackburn and Sachs, 1989). Based on these properties the SBCs can be reliably differentiated from the second type of principal neurons in the AVCN, the stellate cells. By considering these physiological properties, we are confident that the recorded cells were large SBCs, which have been shown to project to the MSO in cats (Osen, 1969). We added this information at the beginning of the Results section.

Each SBC receives between 1-3 endbulb inputs in mice (Cao and Oertel, 2010), and the majority of rostral SBCs seem to receive a single endbulb in cats (Ryugo and Sento, 1991), whereas GBCs are reported to receive between 10-70 endbulb inputs (Liberman, 1991; Spirou et al., 2005). Our dataset and previous studies recording from low-CF SBCs in the Mongolian Gerbil suggest that the majority of cells receive a single functional endbulb input (Kuenzel et al., 2011, 2015). This conclusion is supported by the following findings:

(i) Multiple endbulb inputs to SBCs would be indicated by a larger variation of EPSP slopes reflecting differences in the convergence of different inputs which – during in vivo recording of spontaneous activity – will not be completely synchronized. The consequently expected variations in the shapes (and amplitudes) of EPSPs were not observed in our recordings.

(ii) Multiple endbulb inputs onto SBCs would be indicated by violation of the refractory period and temporal overlap of two or more EPSPs. Our data on EPSP slopes show a unimodal distribution and we did not observe an overlap of multiple EPSPs in the recordings.

(iii) Multiple endbulb inputs would likely be reflected by ANF input rates above the rates of single auditory nerve fibers, since converging ANF inputs on bushy cells originate from the same SR type (Ryugo and Sento, 1991).

To further deal with this issue we compared the distribution of ANF input rates of our recordings with published data from recordings of single AN fibers from the Mongolian Gerbil (Schmiedt, 1989; and Müller, 1996; see Figure 11). We extracted the data from Figure 7 in Schmiedt (1989) and from Figure 3 in Mueller (1996), constructed histograms and compared these to the histogram of ANF input rates of our recordings. In our recordings we did not observe cells with low spontaneous rates, which was expected, since SBCs are predominantly contacted by high-SR ANF (Smith et al., 1993). The distribution of spontaneous ANF input rates observed in the present study matches the distribution of spontaneous rates obtained from single auditory nerve fibers, suggesting that a single ANF is sufficient to produce the observed spontaneous rates. The largest endbulbs were reported at CFs between 1-4 kHz, the CF range of this study (Rouiller et al., 1986; Sento and Ryugo, 1989). Thus, our data are naturally biased towards large SBCs that receive large endbulb endings.

Furthermore, in our recordings we observed an activity-dependent facilitation of EPSPs, which might be caused by residual calcium accumulation in the endbulb of Held (Borst, 2010). If our recordings would contain EPSPs from multiple independent endbulbs, we would not see such facilitation, since only a fraction of EPSPs should be affected by previous activity. Also, the observation of facilitation rather than depression is unlikely to occur with multiple endbulb inputs.

Author response image 2.The experimentally determined ANF input (top) showed spontaneous rates consistent with previously published auditory nerve recordings of the Mongolian Gerbil (middle & bottom; data extracted from Figure 7 in Schmiedt (Hearing Research, 1989) and from Figure 3 in Mueller (Hearing Research, 1996) respectively).**DOI:**
http://dx.doi.org/10.7554/eLife.19295.025

*4) Spike depression. Lorteije et al., 2009 suggested that "spike depression" (presumably the refractory period) contributed to EPSP failures at the calyx of Held. The refractory period would likely appear as an increase in threshold during periods of high activity, similar to Figure 2 right, and a decrease in spike amplitude with high activity, similar to Figure 2 right. This possibility should at least be acknowledged, and the strychnine results interpreted with this in mind as well.*

We agree that spike depression and the influence of refractory period could potentially contribute to failures in AP generation in SBCs. This effect was also addressed in a previous study by Kuenzel and colleagues (2011) who showed that shortly after an SBC spike the failure fraction is increased, similarly to the results reported in the study at hand. Both Lorteije et al., 2009 and Kuenzel et al., 2011 reported short recovery times of 0.8 ms and 2.1 ms, respectively, indicating that the influence of spike depression on signal transmission is limited to periods shortly after the last SBC spike.

We attempt to address the influence of spike depression on AP generation in vivo by recording spontaneous activity in the absence of sound stimulation, when the inhibitory influence is likely small (note that the block of glycinergic inhibition did not influence the input-output function in the absence of sound stimulation, Figure 4).

During acoustic stimulation, the block of glycinergic inhibition counteracts the increase in threshold EPSP, but does not render the synapse fail safe. We do not deny that spike depression will contribute to the input-output function at the ANF-SBC synapse, however argue that this effect is constrained to a short time window after the last spike. Additionally, the observed general increase in threshold EPSP during acoustic stimulation was absent or reduced when blocking glycinergic inhibition, indicating that the effect of spike depression on failure rate is limited. However, we now acknowledge the influence of spike depression more clearly in the Discussion section.

*5) Second paragraph subsection “Synaptic depression alone fails to fully account for increased failure rates”: Mathematically, what you have done here is correct, but it will likely be confusing to many readers. It is unclear why an "error function" (Boltzmann?) is used to fit this data, as the points clearly cluster into two clouds, and the "threshold" point is not near any of the data points. Why were the data not binned along the ordinate of rising slope (in 0.5 or 1 V/s bins), and the probability of success or failure computed in each bin, based on the total number of succ+fail. This would give the type of representation that would then make sense in terms of fitting to a Boltzmann to obtain a "threshold". Maybe do this as an inset?*

We agree with the reviewer that it is intuitively not comprehensible that the current approach is a valid method for the estimation of threshold EPSP. The criticism of the reviewer is well taken and we not only would like to justify the approach in this letter but also explain the approach in in more detail in the text body of the manuscript. Choosing a Gaussian error function was based on a previous study (Kuenzel et al., 2011). Fitting different kinds of sigmoid functions (e.g. Boltzmann function, logistic function) resulted in different slope parameter, but identical inflection points, thus providing the same estimates for the threshold EPSP. In the original submission, we refrained from binning the rising slopes to avoid potential effects of bin size on threshold EPSP estimation. Also, in the method applied in the original manuscript, the fit is inherently weighted by the number of events, which is not the case when using binned data, as all bins will be equally weighted independent of the number of events in each bin, i.e. a probability calculated from e.g. 3 events contributes equally to the fit as does a probability calculated from 1000 events. This is why we thought the use of raw, unbinned data would provide more reliable data. However, we agree that the approach suggested by the reviewers is a more intuitive method of threshold calculation. We therefore recalculated the threshold EPSPs throughout the manuscript using the suggested method (bin size=0.5 V/s). We added a linear weighting parameter to each bin based on the number of events in each bin. The results are virtually identical to the previous approach (Pearson’s r=0.9996, Figure 12). We changed the respective passage in the Methods and Results section and modified Figure 2 accordingly, which now shows the binned fraction of EPSPsucc across the EPSP rising slopes.

Author response image 3.Comparison of threshold EPSP estimation methods: Right: Representative cell.The version in the revised manuscript is based on the probability distribution of EPSP_succ_ as a function of binned EPSP slope (top). The original version was based on a unbinned binary distribution between the EPSP_succ_ and EPSP_fail_ (bottom). Black arrow and red line indicate estimated threshold EPSP. Right: Population data for 62 cells show that both methods provide almost identical estimation for threshold EPSP.**DOI:**
http://dx.doi.org/10.7554/eLife.19295.026

The interchangeability between both types of analysis can be seen in Figure 12 which we would like to share with the reviewers: Both methods have been applied to the representative dataset in Figure 2 (upper left: method using bins, lower left: original method without binning), and the threshold EPSPs have been recalculated for all units in the manuscript (right). Both methods produce virtually identical results (dots close to line of equality, gray).

*6) The summary plots can be very difficult to read. Each plot shows median, quartiles and outliers, as well as averages. This gets very busy, and winds up obscuring or minimizing the important differences. For example, in Figure 3Ciii the averages are squashed down in the lower part of the panel, and most of the panel is taken up with outliers. It would be better to maximize the plot area to make the data as visible as possible. Averages and standard errors are sufficient when testing with a t-test, as are box-and-whisker plots when a non-parametric test. This applies to summary plots throughout, with Figure 5 particularly hard to read.*

The reviewer is correctly pointing out, that the way the data were presented in the graphs made it difficult to single out the important effects. We choose this approach to show all observed values and scaled the axes to cover all theoretically possible values (as in Figure 2, where correlation coefficient is bound between -1 and 1.) We added the marker for the representative cell to transparent show, how its values compare to the rest of the population. In the revised manuscript we followed the reviewers’ suggestion to increase the plot area in the summary plots accordingly. In Figure 3Ciii we added a second y-axis to spread the data for the Q40 values, and in Figure 5 and Figure 6, we replaced the boxplots with markers indicating the mean +/- standard deviation, which makes the graphs much easier to read.

*7) Many plots compare paired values, and report the original values (+/- SE), but not their differences. For example, 3Ci reports average CFs of ANF and SBC. The average values of these are not very meaningful. What is important is the difference between CFs for input vs. output, which is not reported. This applies in many other places: in Figure 3, differences in threshold, Q, asymmetry; in Figure 4 right panels, effects of glycine, strychnine, stimulation; in Figure 5 and Figure 6, differences between input and output; Figure 7, differences between input and output, Figure 8iii, iv, differences between input and output, effects of strychnine.*

We thank the reviewer for pointing out that relevant information was missing the way the data were presented. Our motivation had been to provide a transparent presentation of the distribution of the raw data. We now added mean ± standard deviation for differences in the paired comparison to provide this previously missing information to the reader.

*8) ANOVAs in Figure 5, 6. It is very difficult to understand the ANOVA carried out here. In Figure 5, left, the ANOVA results suggest there is "a systematic variation with modulation frequencies". This is not visible in the plot, and furthermore, the nature of the "systematic variation" is obscured by an ANOVA. Did it go up or down? In the FM analysis in Figure 6, the extreme frequencies show little effect, but 100 Hz modulation shows a strong effect. Why is this behavior so non-linear? Did the results match some sort of expected outcome?*

We apologize for the misleading term “systematic variation” when describing the results of the VS across stimulation frequencies. Our focus was on the comparison between ANF input and SBC output rather than changes across modulation frequencies. Regarding the VS values, for both ANF input and SBC output the VS decreased with increasing stimulation frequency (VS 50 Hz: ANF=0.26 ± 0.09 vs. SBC=0.31 ± 0.07: 400 Hz: ANF=0.2 ± 0.1 vs. SBC=0.25 ± 0.1). While relatively stable VS values are obtained up to 200 Hz modulation frequency, the VS was significantly reduced at 400 Hz for both ANF input and SBC output. Importantly, the VS of the SBC output was larger than the ANF input across all frequencies tested.

The reviewer correctly observed the effect of the modulation frequencies, and here is, how we would like to explain this finding: The non-linearity of the frequency effect could be explained by the time-constant of the impact of inhibition. The FM stimulus spanned 3 octaves around the unit’s CF, thus covering frequency areas with strong inhibition and areas lacking any inhibitory effect. The time constant of the inhibitory effect is around 20-50 ms in slice recordings (Nerlich et al., 2014), but somewhat shorter in vivo (10–15 ms. Nerlich et al., 2014; Keine and Rübsamen, 2015). During frequency modulated stimulation, the inhibitory conductance should therefore also be modulated and essentially act as a band-pass filter. The slowest and highest modulation frequencies might result in saturation of inhibition or insufficient activation (if the inhibitory conductance does not summate), thus reducing the inhibitory effect on the ANF-to-SBC synapse. However, the exact inhibitory dynamics in vivo are not well understood, also the combination of cell types exerting this inhibition on SBC is unclear. Hence, the temporal dynamics of inhibitory inputs during acoustic stimulation remain elusive. At this point, we can only speculate, why the effect is smaller at the highest and lowest frequencies tested, but it is tempting to argue, that the inhibitory dynamics capture the modulation rates of natural acoustic stimuli. Further studies employing natural sounds would be necessary to evaluate the inhibitory effect on different time scales.

*9) Flatness of the RLF (Figure 3). Using standard deviation to quantify flatness of the RLF is confusing. A standard deviation is a comparison to a mean, and the mean here is not meaningful because it depends on which intensities were sampled. My sense is that the standard deviation would depend on whether the RLF is sampled symmetrically about a cell's dynamic range (the SD would be maximal), or asymmetrically (the SD would be lower). Please justify this measure more carefully. A more intuitive measure of flatness may be the difference in firing rate between maximum and minimum (normalized appropriately).*

The reviewer is right in pointing out, that the measure of rate-level variability (RLV) as the standard deviation of the rate-level function (RLF) will depend on the intensities that actually were sampled. We calculated the standard deviation across all of the sampled intensities, identically for the ANF input and the SBC output. In most cells, the RLF of the ANF input shows monotonic increase over a relative wide range of firing rates, while the RLFs of the output were more flat or even non-monotonic. A measure of monotonicity was previously suggested by Kuenzel and colleauges, (2015), which relates the maximum firing rate throughout the RLF to the firing rate obtained at highest sound intensities. While this measure is suitable for detecting non-monotonic behavior in the SBC RLF, it does not quantify the gain of neuronal response, i.e. the change in firing rate across the sound intensity. Since non-monotonic behavior of the RLF is only present in about half of the SBC (Winter and Palmer, 1990; Kopp-Scheinpflug et al., 2002; Keine and Rübsamen, 2015), we opted to use another measure. The original measure was normalized to the spontaneous activity and sampled identically for all cells, and thus quantifies the deviation of firing rates relative to the spontaneous activity. Still, the reviewer is right in pointing out that in the original analysis, the RLV will depend on the threshold of the neuron, i.e. for identical changes in firing rates, different measures of RLV might be obtained. While this should not pose a major problem for the comparison of ANF input and SBC output (which are both driven by the same ANF), we agree that the way the data were analyzed might be somewhat puzzling for the reader. We therefore calculated the RLV as suggested by the reviewer as the difference between minimal and maximal firing rate normalized to the spontaneous activity of ANF input and SBC output, thus providing a measure of the change in firing rates relative to spontaneous activity independent of the sampled sound intensities or the unit’s threshold. Consequently, the RLV was renamed rate-level gain (RLG) throughout the manuscript. The new analysis is shown in the revised Figure 3. Although both measures quantify somewhat different properties of the RLF, a comparison between both resulted in significant positive correlation between the two (input: rs = 0.36, p < 0.01, output: rs = 0.35, p < 0.01). Therefore, using the method suggested by the reviewers allows us to draw the same conclusion about the reduction in gain, but is more intuitive to the reader.

*10) Effects of inhibition. The analysis of the effects of inhibition is very thorough, and beautifully shown, but does not adequately acknowledge similar results from earlier work. This is particularly noticeable in the final paragraph of subsection “Acoustically evoked inhibition elevates threshold for AP generation”, which does not acknowledge that Kopp-Scheinpflug, 2002 and Kuenzel, 2011 also showed similar tuning of inhibition. Figure 5–Figure 7 investigate different types of stimuli, but it would be helpful to compare the findings against others' results using tuning curves. The period histograms in Figure 5 and Figure 6 seem to show that the output is simply shifted lower compared to input. Is such a simple overall shift sufficient to explain the changes in vector strength, modulation depth, reproduction, and correlation? Or, are the temporal characteristics of inhibition also important?*

As the reviewer correctly pointed out, the analysis of tuning curves showed similar characteristics of inhibition as reported earlier. While we did not observe two distinct types of inhibition, consistent with Kuenzel et al. (2011), we found inhibition to be broader than excitation, which is consistent with (Caspary et al., 1994; Kopp-Scheinpflug et al., 2002). We now acknowledge and cite these earlier findings now in the revised manuscript.

One of the main functions of inhibition seems indeed to be the control of gain, i.e. an overall shift in the firing rates and less dependence of firing rate on sound intensity. However, various studies showed, that the temporal precision also improves from the ANF to SBC on a single-unit level (Dehmel et al., 2010; Kuenzel et al., 2011; Keine and Rübsamen, 2015). Recent studies analyzing the EPSP size showed, that large EPSPs are better timed than small EPSPs (Kuenzel et al., 2011). Modeling studies showed that an increase in temporal precision can be achieved by a sufficiently large number of converging inputs (Rothman et al., 1993; Kuenzel et al., 2015) and such non-endbulb inputs have been reported in the rat VCN (Gómez-Nieto and Rubio, 2009). A recent modeling study showed that inhibition in combination with excitatory inputs at the dendrites could increase the temporal precision, but inhibition alone cannot (Kuenzel et al., 2015). Additionally, as documented by Nerlich et al. 2014 (and already mentioned above) the inhibitory inputs are rather slow, thus not suited to act on a spike-to-spike basis.

To answer the question of the reviewer, we tested if the increase in vector strength, temporal precision, reproducibility and reproduction can be achieved by different mechanisms of inhibitory action. First, we simulated a pure divisive effect of inhibition, i.e. a random loss in output spikes. For that we used the input spike times of our recordings and pseudorandomly removed spike times to match the failure fractions observed in our recordings. The simulated SBC output then reflects a pure (divisive) scaling of the ANF input with failure rates identical to the experimental data. This simulation showed that a random reduction in postsynaptic firing rates is not sufficient to cause the observed increase in temporal precision and trial-to-trial reproducibility neither for SAM nor SFM stimulation (see Figure 9—figure supplement 1 and Figure 9—figure supplement 2). Next, we simulated a purely subtractive effect of inhibition by removing a fixed number of output spikes per bin, again matched to the experimentally observed failure rates. This results in an overall shift of the histogram, without affecting the gain of the response. We observed an increase in vector strength, modulation depth, reproducibility and sparsity for both the SAM and SFM stimulation. Notably, the simulated increase during SAM stimulation even exceeded the experimental data, while during SFM stimulation, the simulated subtractive inhibitory effect matched the data well.

Finally, we applied the same simulations to the data obtained during RGS stimulation. Again, we find that purely divisive inhibition cannot explain the increase in sparsity, reproducibility and temporal precision, while pure subtractive inhibition matches the data well. The simulation of purely subtractive inhibition shows a larger improvement in sparsity and reproducibility than observed in the data.

We conclude that the subtractive shift in SBC output rates compared to the ANF input can explain the increase in most metrics. A subtractive effect on the firing rates has been associated with hyperpolarizing inhibition with slow temporal dynamics (Doiron et al., 2001), consistent with previous slice recordings in SBCs (Nerlich et al., 2014). However, we also observed a reduced response gain of the SBC output, which is generally attributed to shunting inhibition (Rose, 1977; Koch and Poggio, 1992). Previous studies showed that both types of inhibition might be present at the SBC synapse and our data suggest that these two mechanisms result in a two-fold effect of increased temporal precision and reduced sound level dependence. While during SFM stimulation with constant stimulus level, the simulated subtractive inhibition matches the observed data well, during SAM stimulation, the simulated subtractive inhibition overestimates the effect on the SBC output. This might be caused by the inherent changes in stimulus level and the additional effect of inhibition on response gain in the experimental data.

Regarding the temporal dynamics of inhibition: Considering the slow dynamics of inhibitory currents, the temporal relation between inhibition and excitation might play only a minor role. It seems conceivable that the slow inhibitory currents will summate during acoustic stimulation, resulting in a functionally tonic inhibitory conductance (Nerlich et al., 2014; Kuenzel et al., 2015). However, it has been shown, that well-timed EPSPs are more likely to generate an SBC output than poorly timed EPSPs. Also, experimental data provided evidence that indeed the EPSP size, and – depending on this – the probability of SBC AP generation, does partially depend on the timing of the event (Keine and Rübsamen, 2015; Kuenzel et al., 2015), and it has been suggested that non-endbulb excitatory SBC inputs might facilitate AP generation of well-timed endbulb EPSPs (Ryugo and Sento, 1991; Gómez-Nieto and Rubio, 2009). The degree of non-endbulb excitatory inputs converging on SBCs and their activation pattern during acoustic stimulation remains to be investigated to draw conclusions of their importance of signal processing.

*11) FM data period histograms. The FM stimuli appear to yield 2 or 3 peaks in the representative example of Figure 6. Multi-peaked firing can undermine the utility of the standard VS calculation. That is, each peak may be very tight and repeatable, but by having more than one, the VS decreases.*

We agree with the reviewer that using VS for distributions with multiple peaks does not provide a sufficient measure of the temporal precision or reproducibility of the neuronal response. We included the VS for SFM stimuli to use a traditional, standardized measure for comparison with the SAM data and previous publications. We complemented VS with other measures of modulation depth (which for example deals well with multiple peaks), reproducibility and CorrNorm to provide a more comprehensive description of the temporal response properties. Also, the main focus of this analysis targeted a direct comparison of ANF input and SBC output rather than changes across modulation frequencies.

However, to inform the reader about the potential problems in VS interpretation for the observed phase histograms, we added a paragraph at the Results section which reads:

“It has to be considered that the interpretation of VS values is difficult when the phase histogram of the neuronal response shows multiple peaks (Figure 6). We, therefore, used a set of additional measures to describe the neuronal response to SFM stimuli when comparing ANF input and SBC output.“

*12) STRF analysis: This analysis is very nice, but some clarification would be helpful.a) Why is the inhibitory STRF calculated by subtracting the ANF and SBC plots? Please explain how and why firing rates are normalized by the standard deviation. Is it normalized by subtraction or division? What if this normalization is not done? Please add a scale bar to these plots. Can this analysis be done instead by using only the EPSP(fail) for the kernel estimation? How would this approach differ from the subtraction approach?b) The size of the excitatory region is described as sharper in the SBC compared to ANF. This is not obvious in the example. Can the spectral width be indicated on the plot in Figure 7 somehow? Is there a similar effect in the tuning curves in Figure 3? It is not clear how sound level influences the STRF. Would that affect this width measurement?*

We are happy to provide the information the reviewer asked for. We realize that some necessary information was missing in the original manuscript which happened by trying to keep the text succinct.

Strictly speaking, we computed the difference STRF by subtracting the STRFs of ANF from the STRF of the SBC. Basically, this difference STRF could have positive and negative parts, but – in the recorded datasets – was dominated by negative parts, i.e. indicative of a reduction in response, which could be described as 'inhibitory STRF' (i.e. wherever the SBC STRF is smaller than the ANF STRF, some spikes were missing, and this could be due to inhibition), although not all missing spikes needed to be an indication for inhibition. We therefore (as far as we know) do not refer to it as "inhibitory STRF", but more generally as difference-STRF.

For us, the normalization seems to be an appropriate procedure, since otherwise, the change in gain (i.e. overall firing rate which in part was also due to unspecific failures of spike transmission) dominates the change in shape of the STRF. We originally computed it on non-normalized STRFs, which leads to a similar result after the excitatory peak. However, one obtains a dominating, large peak at BF x Best Latency, which visually masks the changes thereafter. Figure 13 documents the observed difference without normalization. The units of the STRF are Hz/dB SPL, as they map from the dB scaled spectrogram to firing rate.

Author response image 4.**DOI:**
http://dx.doi.org/10.7554/eLife.19295.027

As suggested by the reviewer, we added color scale bars to two of the plots in the revised Figure 7 (top two plots are on the same scale, normalization is performed according to the standard deviation of the STRF, and then commonly normalized to peak close to 1).

The difference STRF could also be computed using the EPSPfail, which would mathematically be equivalent (if the appropriate weighting is included) to the difference without normalization. We chose to perform it as the difference since it offers the possibility to normalize (essentially to focus on the shape, rather than overall rate) in between the responses, and we can visualize each of the STRFs in comparison.

Regarding spectral width: Indeed, the spectral half-width is only slightly reduced, and this is not the case not in all units. We now indicate the spectral half-width in the example cell visually, demonstrating that the unit shown is a representative example (see also subpanel E).

The potential influence of sound level on the shape of the units’ response areas is an interesting suggestion: It would be interesting to test the level dependence of ANF and SBC output separately with the prediction that output shape would be more constant than input shape. Given the inhibitory influence, this could hold up to high sound levels, but may deteriorate at low sound levels.

In previous studies in the MNTB (Englitz et al., 2010), we experimented with different choices of sound level, essentially finding similar results as (Rabinowitz et al., 2011) in the auditory cortex (testing for sound contrast processing), namely that the shape of the STRF is mostly invariant, but that the scaling (e.g. via an output nonlinearity) is changed. Therefore, it could also be that changes in gain would be hidden in the STRF structure. But, at this point, this is speculation since in our stimuli overall loudness was not modulated. A consideration of possible sound level influence has been added to the Discussion.

The computational aspects regarding the benefit of normalization are now explained in more detail in the Methods section, right where the STRF is introduced, and furthermore are also briefly mentioned in the Results section (with reference to the Methods).

13) Figure 8: This is a very important figure, because it addresses how inhibition influences the information carried by SBCs compared to ANFs. It would be very helpful if the authors could take a stab at explaining how inhibition may enhance sparsity, reproduction, and temporal dispersion. In the case of sparsity, it seems like it cannot be simply a change in the mean firing rate, because kurtosis should not include that (unless there are weird effects of rectification at firing rate of 0).

Thank you for encouraging us to address this topic directly: since reviewer point #10 asked a similar question, we partially have dealt with the mechanism above.

We think that the main effect of inhibition is a general shift of the response rates closer to quiescence (due to combined divisive/subtractive effects). This shift is triggered by sound evoked glycinergic input, which covaries with excitation, but arrives slightly delayed (~3ms, Keine and Rübsamen, 2015)http://f1000.com/work/citation?ids=1980015. It prevents ANF inputs from evoking SBC output spikes, a mechanism especially affecting poorly timed endbulb inputs (Kuenzel et al., 2011, 2015; Keine and Rübsamen, 2015). As a consequence, sparsity increases, as in many cases mainly the previously high peaks in firing rate remain, whereas a bulk of PSTH bins is shifted to low or zero firing rate bins. As indicated by the reviewer, this will lead kurtosis to increase, as well as the other two measures of sparsity (see Figure 8—figure supplement 1 for a comparison of sparsity).

More generally, we think that the subtractive part of the glycinergic inhibition, in combination with the well-timed, high firing rate events, can explain the improvement of sparsity, reproducibility and temporal precision. We tested this hypothesis by simulating the effects of simple divisive (remove fixed fraction of spikes – relative to the firing rate – from a time bin in a spiketrain) and subtractive (remove fixed number of spikes from each time bin in a spiketrain) inhibition (see Figure 9). The removal of events was matched to the experimentally observed failure fractions. It appears the subtractive part of the inhibition can achieve these improvements, whereas the divisive part leaves them unchanged. On the other hand, the divisive part, together with the co-tuning of excitation and inhibition may preserve the SBC’s ability to respond in a limited range of firing rates (Figure 3Biii/D). It should be noted that the simulated subtractive inhibition resulted in greater improvements than observed in the experimental data, additionally supporting a combined subtractive/divisive inhibitory action. A further, theoretical exploration of this combination appears beyond the scope of this manuscript.

From the literature it appears that the effect of glycinergic inhibition is not clearly subtractive or divisive (Kuenzel et al., 2011, 2015; Nerlich et al., 2014). Hence, both a general increase in membrane permeability (leading to divisive firing rate effects) as well as shifts of membrane potential by hyperpolarization (during a phase of depolarization, leading to subtractive effects on the firing rate level) seem to shape the SBC output (see Ayaz and Chance (2009) for more details). Subtractive and divisive inhibition have been shown to differently affect the neuron’s output and tuning properties. While sole subtractive inhibition can sharpen the tuning and shifts the stimulus-response function without altering the gain, divisive inhibition preserves the tuning and influences the response gain (Wilson et al., 2012).

We think that such a – potentially temporally unspecific – effect is nonetheless compatible with an increase in temporal precision, if the high peaks of the ANF firing rate are linked to specific acoustic events. In the cross-correlations underlying the computations of reproducibility and temporal precision, a removal of the bulk of spikes, and a focus on the peak rates, would lead to higher, tighter correlations. These precisely timed spikes are then preserved and transmitted to the MSO (see #1 for more detail). Assuming that the same occurs at the other AVCN, this would hence lead to a sharpening of interaural temporal correlation.

These arguments were added to the Discussion of the revised manuscript as a new paragraph “Inhibitory mechanism for improving sparsity and reproducibility of the neural response”